

# Observation and modelling of snow at a polygonal tundra permafrost site: spatial variability and thermal implications

Isabelle Gouttevin[1,2], Moritz Langer[3,4], Henning Löwe[5], Julia Boike[3,4], Martin Proksch[5], and Martin Schneebeli[5]

[1]Irstea, UR HHLY, centre de Lyon-Villeurbanne, 5 rue de la Doua, BP 32108, 69616 Villeurbanne Cedex, France.
[2]Université Grenoble Alpes, Irstea, UR ETGR, Centre de Grenoble, 2 rue de la Papeterie-BP 76, 38402 St-Martin-d'Hères, France.
[3]Alfred Wegener Institute, Helmholtz Center for Polar and Marine Research (AWI), Telegrafenberg A6, 14473 Potsdam, Germany.
[4]Department of Geography, Humboldt-Universität zu Berlin, Rudower Chaussee 16, 12489 Berlin.
[5]WSL Institute for Snow and Avalanche Research SLF, Flueelastr. 11, 7260 Davos Dorf, Switzerland.

*Correspondence to*: Isabelle Gouttevin (isabelle.gouttevin@gmail.com)

**Abstract.** The shortage of information on snow properties in high latitudes places a major limitation on permafrost and more generally climate modelling. A dedicated field program was therefore carried out to investigate snow properties and

their spatial variability at a polygonal tundra permafrost site. Notably, snow samples were analysed for surface-normal thermal conductivity ($K_{eff-z}$) based on X-ray microtomography. Also, the detailed snow model SNOWPACK was adapted to these Arctic conditions to enable relevant simulations of the ground thermal regime. Finally, the sensitivity of soil temperatures to snow spatial variability was analysed.

Our depth hoar samples were found more conductive ($K_{eff-z}$ = 0.22 ± 0.05 W m$^{-1}$ K$^{-1}$) than in most previously published

studies, which could be explained by their high density and anisotropy. Spatial variations in the thermal properties of the snowpack were well explained the micro-topography and ground surface conditions of the polygonal tundra, which control depth hoar growth and snow accumulation. Our adaptations to SNOWPACK, phenomenologically taking into account the effects of wind compaction, basal vegetation and water vapour flux, yielded realistic density and $K_{eff-z}$ profiles that greatly improved simulations of the ground thermal regime. The potential of an anisotropy and density-based formulation of $K_{eff-z}$ in

snow models was shown. Soil temperatures were found to be particularly sensitive to snow conditions during the dark part of winter, highlighting the need for improved snow characterization and modelling over this period.

## 1 Introduction

Perennially frozen ground (permafrost) is a major feature of high-latitude regions, underlying about 25 % of the northern hemisphere. This essential climate variable reacts sensitively to ongoing climate change, with important implications for

terrain stability, coastal erosion, surface and subsurface water fluxes, the carbon cycle, and vegetation development (e.g. Grosse et al., 2016; Shuur et al., 2015). Understanding and modelling the thermal regime of permafrost is therefore essential





for a broad variety of applications ranging from geo-engineering to landscape preservation and climatic projections, and also for ecological considerations.

The influence of snow cover on the ground thermal regime has been highlighted by a number of authors (e.g. Sturm and Holmgren, 1994; Zhang et al., 1996; Zhang, 2005; Lawrence and Slater, 2010; Gouttevin et al., 2012; Langer et al., 2013;

Dominé et al., 2015, 2016a, 2016b). Snow has a low thermal conductivity ($K_{eff}$), ranging from 0.01 to 0.7 W m$^{-1}$ K$^{-1}$ depending on microstructure, density and wetness, and it therefore insulates the underlying ground during the cold season. The soil temperatures beneath a thick snowpack will therefore be warmer than under a thin snowpack (or no snowpack at all), given similar meteorological conditions.

Arctic tundra regions are usually characterized by thin but enduring snowpacks. At the Samoylov permafrost observatory

(Lena River Delta, Siberia, 72° N, 126° E), snow covers the ground for on average 7 months of the year, with the mean February snow depth ranging between 15 and 30 cm (Langer et al., 2013). Under such conditions (long duration of the snow cover and thin snowpack) the sensitivity of the ground thermal regime to the surface-normal snow thermal conductivity $K_{eff\text{-}z}$ is particularly high (Zhang, 2005). An extensive investigation by Langer et al. (2013) into the sensitivity of the ground thermal regime at Samoylov showed that the thermal properties of the snow were the most essential parameters to constrain

for accurate simulation of the permafrost thermal regime.

The insulating power of snow on the underlying ground is linked to the surface-normal component of the conductivity tensor $K_{eff\text{-}z}$ and to the height of snowpack $HS$. It can be expressed as the thermal resistance ($R_{th}$), where $R_{th} = HS / K_{eff\text{-}z}$. Assessing the $K_{eff\text{-}z}$ of a natural snowpack is not easy. It is often estimated in situ with the help of a needle-probe (NP) inserted in the snow parallel to the surface, which actually allows to estimate $\sqrt{K_{eff-z}K_{eff-x}}$, i.e. a combination of the surface-normal ($K_{eff\text{-}z}$)

and parallel ($K_{eff\text{-}x}$) components of $K_{eff}$ (Riche and Schneebeli, 2013). Since most snow types are anisotropic with regard to $K_{eff}$ (meaning that $K_{eff\text{-}z}$ is not equal to $K_{eff\text{-}x}$; Riche and Schneebeli, 2013), a correction for anisotropy needs to be applied in order to obtain $K_{eff\text{-}z}$ from an NP measurement. Snow samples also can be analysed for $K_{eff\text{-}z}$ in cold laboratories, either using a guarded heat-flux plate (HFP), or by combining X-ray microtomography with direct numerical simulations at a microstructural level (CT). The differences between these three measurement techniques have been investigated by Riche

and Schneebeli (2013), who found that NP estimates were on average 35 % lower than CT estimates, even after correcting for anisotropy. While HFPs tended to yield higher estimates of $K_{eff\text{-}z}$ than CT, the difference was smaller than with NP (20 % on average) and could reasonably be ascribed to identified uncertainties in the HFP and CT methods. After improving their NP $K_{eff}$ retrieval algorithm and taking anisotropy into account, Dominé et al. (2015) reassessed the systematic residual difference between NP measurements and the CT results to about 20 %. However, an additional complication occurs when

an NP is used in depth hoar (a columnar snow type frequently encountered in the lower part of Arctic snowpacks): apart from being highly anisotropic, the fragile structure of depth hoar can be damaged during needle insertion, reducing the quality of the measurements. The only depth hoar sample considered in the methodological comparison by Riche and Schneebeli (2013) exhibited the largest difference (55 %) between anisotropy-corrected NP measurements and CT estimates, probably as a result of these limitations. Overall, the CT method currently seems to provide the most reliable estimates for



$K_{eff\text{-}z}$. However, the constraints of casting and transporting samples for cold-laboratory analysis reduce its applicability for continuous monitoring and for investigations at remote sites. Almost all present-day $K_{eff\text{-}z}$ estimates for Arctic snowpacks are therefore based on NP measurements (Barrere et al., 2017, Dominé et al., 2016b).

Statistical models for $K_{eff}$ or $K_{eff\text{-}z}$ (mainly as functions of density) have been developed to provide this parameter to snow and

permafrost models in the absence of observational data. Such density-based regressions are inherently only able to account for parts of the variations in $K_{eff\text{-}z}$, as the development of some snow types (such as depth hoar) is accompanied by changes in their microstructural anisotropy that affect the $K_{eff\text{-}z}$ even if the density remains unchanged (Löwe et al., 2013; Calonne et al., 2014). Although regressions that include the effect of anisotropy have been established (Löwe et al., 2013), they require additional input in the form of an anisotropy parameter.

Most of the current generation of detailed snow models such as CROCUS (Vionnet et al., 2012) or SNOWPACK (Bartelt and Lehning, 2002; Lehning et al., 2002a, 2002b) rely solely on density to infer $K_{eff\text{-}z}$. However, these models are unable to reproduce the density profiles actually observed in Arctic snowpacks (Barrere et al., 2017, Dominé et al., 2016a), which has an immediate impact on the inferred value of $K_{eff\text{-}z}$. A first probable cause of this failure is that these models do not represent the upward water vapour flux, that redistributes ice from the bottom of the snowpack to the upper part as a result of steep

temperature gradients. Dominé et al. (2016b) have estimated that this process could lead to density changes up to 100 kg m$^{-3}$. Additional uncertainties occur in these models in their representation of wind-induced compaction (Groot-Zwaaftink et al., 2013) and the effect of low or basal vegetation (dwarf shrubs, sedges) on snow compaction and metamorphism (Dominé et al., 2015). Intertwined twigs within the snowpack can promote depth hoar formation by preserving an aerated layer, protected from wind erosion and compaction, where conductivity is weak and steep temperature gradients can establish,

favouring rapid metamorphism (Hutchinson, 1965, Sturm and Benson, 1997). The warming effect of protruding twigs in early winter may also enhance snow metamorphism (Sturm and Holmgren, 1994).

The insulating power of snow depends not only on $K_{eff\text{-}z}$ but also on snow height HS. Arctic and high-Arctic permafrost regions such as Samoylov commonly feature polygonal tundra landscapes, which are characterized by a distinctive micro-topography with polygons that are typically about 10 m wide and rims that are about 1 m high. This micro-topography

induces considerable variations in snow depth (Wainwright et al., 2017), with significant implications for the functioning of the local ecosystem including the thermal regime, hydrology, and carbon cycle (Liljedahl et al., 2016; Hobbie et al., 2000). Thus, an integral assessment of snow thermal conductivity, snow depth and their spatial variability, is needed to fully characterize the thermal impact of snow on permafrost in polygonal tundra landscapes.

Our objectives in this study were (1) to acquire information on the thermal properties of snow using the CT technique, and

analyse their spatial variability in relation to the polygonal tundra micro-topography (2) to phenomenologically adapt the SNOWPACK snow model to the Samoylov context based on current knowledge of Arctic snow processes, and carry out consistent numerical simulations of the ground thermal regime at a polygon center, and (3) to estimate the thermal implications of snow spatial variability (in both depth and structure) across a polygonal tundra landscape. To this end we





relied on snow samples and a variety of in situ snow observations collected during a dedicated field program at Samoylov in April 2013, as well as more long-term observations on meteorology and soil variables.

The manuscript is organized as follows: Section 2 provides a description of the data and the methods used. We then present in section 3 our estimates of snow thermal properties and analyse their spatial variations across the polygonal landforms. In section 4 the adaptations of the SNOWPACK model are described. The simulated snow profiles and ground thermal regime at a polygon center are compared to in situ data in section 5: for the simulation of soil temperatures, the modified SNOWPACK is used in combination with the CryoGrid3 (CG3) permafrost model (Westermann et al., 2016) which was extensively validated at Samoylov. Ensemble estimations of the snow thermal properties across a polygon's transect, together with soil temperature records in different micro-topographic conditions, are then used in section 6 to assess the thermal impact of snow spatial variability. Finally, we conclude the paper by comparing in section 7 our findings with those reported in previous publications and highlighting the sensitivities and remaining limitations of snow models in Arctic environments.

## 2 Data and methods

### 2.1 Samoylov site

The Samoylov permafrost observatory is located within the zone of continuous permafrost, on Samoylov Island in the Lena River Delta, Siberia (72° N, 126° E; Fig. 1). The site has been used for intensive monitoring of ground temperatures and meteorological conditions since 1998 (Boike et al., 2013). The mean annual air temperature is -12.5 °C, with mean monthly temperatures ranging from -33 °C to 8.5 °C (1998-2011). The average annual rainfall is 125 mm, while snowfall averages 40 mm yr$^{-1}$. The landscape is characterized by polygonal tundra, i.e. a complex mosaic of dry polygonal ridges with wet depressed centers, and a number of larger water bodies (Muster et al., 2012; 2013).

In the present study we analysed the snow properties with respect to the micro-topography and surface conditions (water-logged, grass-covered, etc.) of the polygonal tundra. We divided the micro-topography into polygon rims, slopes, and depressed centers, referred to simply as rims, slopes, and centers. With regard to the surface conditions, the elevated rims and slopes are usually vegetated (mosses and Dryas species, ~ 20 cm high) while the polygon centers are typically either damp or water-logged. The damp centers are vegetated, mainly with mosses and Carex species (~ 15 to 20 cm high) and are referred to as "grass-centers" while the water-logged centers lie below the water table and are referred to as "ice-centers". The ponded water in these ice-centers forms an ice base beneath the snow cover in winter and spring, which is clearly distinguishable from the moss-grass-snow interface of the 'grass-centers'. We therefore ended up with four micro-topographic classes summarizing the typical micro-topography and surface conditions at Samoylov: grass-centers, ice-centers, rims, and slopes.

During the winter the grasses of the rims, slopes and grass-centers tend to be flattened by snow and in places become intertwined at the base of the snowpack, up to a height of 7 to 10 cm (Fig. 1 d).



## 2.2 Snow data

### 2.2.1 In situ snow observations

The Samoylov snow campaign in April 2013 (Fig. 1) focused on sampling the four afore-mentioned micro-topographic classes in polygons located close to, but not influenced by the Samoylov station. Sixteen stratigraphic profiles were carried

out, with records of grain type, size, and occasionally density, hand hardness, and temperature measurements. Snow samples were cast with diethyl-phtalate, as detailed in Heggli et al. (2009), and were later analysed in the SLF-Davos cold laboratory by CT (Coleou et al., 2001; Schneebeli and Sokratov, 2004). Four sets of samples that covered the stratigraphy of distinct ice-center, grass-center, rim, and slope profiles, were selected for our investigations on the basis of sample integrity. The corresponding sites will be referred to as CT sites (consisting of CT rim site, CT slope site, etc..). An east-west trench was

excavated across a grass-center polygon, which will be referred to as the "reference polygon" due to its denser instrumentation (Fig. 1). Near-infrared (NIR) images of the trench were realized to characterize the thickness of the basal depth hoar (DH) layer along the transect. Snow depth was recorded continuously over the 2012-2013 snow season by an SR50 sensor (Campbell Scientific, ± 1 cm accuracy, ± 1 cm precision) located in the topographically low center of the reference polygon (Fig. 1). However, this snow depth record differed from data acquired at grass-center snowpits: on 21

April 2013 the SR50 measured 13 cm of snow while both the transect and snowpit data indicated depths in excess of 17 cm for grass-center conditions. This difference is likely due to small scale variability in the micro-topography and in processes such as wind erosion immediately below the SR50 sensor. We matched the SR50 snow data to the median of manually recorded snow depths at grass-center snowpits (20 cm) on 21 April 2013, by multiplying the SR50 record by a constant factor of 1.6. Finally, a time-lapse camera provided daily, low-resolution images of the reference polygon.

### 2.2.2 Laboratory analysis

The samples cast in the field were transported to the cold laboratory in Davos and analysed by X-ray microtomography, thereby obtaining 3-dimensional images of the structure and bonding of the ice crystals. Binary micro-tomographic images were used as input for a finite element analysis to calculate the 3-dimensional heat conduction through the porous ice-air medium and thus obtain the effective conductivity tensor for the analysed sample. This conductivity only takes into account

pure conduction through the ice-air network, ignoring the effects of water vapour flux and latent heat. For the heat conductivity calculations we used the procedure described in Löwe et al. (2013), based on NIST Finite Element programs (Garboczi, 1998), with an air conductivity ($k_a$) equal to 0.024 W m$^{-1}$ K$^{-1}$ and an ice conductivity ($k_i$) equal to 2.43 W m$^{-1}$ K$^{-1}$. These figures represent the approximate air and ice conductivities at temperature of -10 °C (cf. engineering toolbox.com and data compiled by Waite et al., 2006).



## 2.3 Soil temperature data

Soil temperatures were recorded over the 2012-2013 snow season from three profiles within the reference polygon (rim, slope, and grass-center) at depths 5 cm, 20 cm and 40 cm, using thermistors (Temperature Probe model 107, Campbell Scientific Ltd., UK). The thermistors were calibrated at 0 °C so that the absolute error was less than 0.1 K over a temperature range of ± 30 °C.

## 2.4 Meteorological data

The SNOWPACK and CryoGrid3 models require as input the following meteorological data: 2 m air temperature, incoming shortwave and longwave radiation, wind-speed, and relative humidity of the air. We drive the models with snow depth recorded by the SR50 sensor instead of precipitation. Air temperature and relative humidity were recorded at the Samoylov meteorological station using an HMP45C air temperature and humidity sensor (Fig. 1). Unfortunately the sensor became saturated at temperatures below -40 °C and so for the period between 1 February and 15 March 2013, when the air temperatures were below -40 °C, we used air temperature records from the ERA-interim reanalysis (ERA-i; Dee et al., 2011) instead. The incoming shortwave and longwave radiation and the wind-speed were also taken from ERA-i as none of these variables was recorded at Samoylov during the 2012-2013 snow season. A comparison of ERA-i with locally acquired meteorological data from other years revealed the ERA-i data to be suitable for the Samoylov site. Snow depth data, meteorological data, and data on the ground thermal conditions at Samoylov during the 2012-2013 snow season are presented in Fig. 2. Meteorological and snow depth data are freely available at https://doi.org/10.1594/PANGAEA.879341.

## 2.5 SNOWPACK snow model

SNOWPACK is a one-dimensional, physically-based snow-cover model. Driven by standard meteorological observations (see Meteorological data), the model simulates the stratigraphy, microstructure, metamorphism, temperature distribution, and settlement of snow, as well as surface energy exchange and mass balance. Snow is represented by a number of state variables (temperature, density, and water content) and the snow micro-structure by grain characteristics (grain size, size of bonds, sphericity, and dendricity). The equations governing the evolution of the seasonal snowpack are described in Bartelt and Lehning (2002) and Lehning et al. (2002a, b), along with the parameterizations adopted for important snow properties, such as $K_{eff-z}$. The latter is based on the work of Adams and Sato (1993), who considered the geometrical arrangement of spherical ice grains to derive an analytical formulation for $K_{eff-z}$. The thermal effect of water vapour diffusion within grain interstices and the temperature dependence of ice conductivity are also taken into account in the parameterization currently used in SNOWPACK. A shape factor calibrated with alpine snow is used to take into consideration the non-sphericity of the snow grains. The SNOWPACK formulation for $K_{eff-z}$ depends in the end on three variables: temperature, density and the ratio between grain-size and bond-size.



The SNOWPACK model was originally developed for alpine conditions (Lehning and Fierz, 2008) but has been recently adapted to different snow and meteorological conditions at the instance of the extreme conditions of the Antarctic Plateau at Dome C: the latter required a specific treatment of the effects of high wind speeds and low temperatures on snow accumulation, compaction and settlement (Groot-Zwaaftink et al., 2013).

**2.7 CryoGrid3 permafrost model**

CryoGrid3 (CG3, Westermann et al., 2016) is a one-dimensional permafrost-soil model that has been extensively adapted and validated for the Samoylov conditions (Westermann et al., 2016; Langer et al., 2016). Since the soil scheme in SNOWPACK lacks the detail and performance of CG3, we used CG3 to model the ground thermal regime but using the snow characteristics (density, depth, and bulk thermal conductivity) produced by SNOWPACK as input.

CG3 is forced by standard meteorological variables (see Section 2.4: Meteorological data) which drive an explicit surface energy balance scheme that simulates the exchange of heat and water with the atmosphere. The model includes a transient heat transfer scheme for the soil that is specifically optimized for simulating freeze-thaw processes within permafrost. The soil physical properties such as heat capacity, thermal conductivity, and the freeze curve, are derived according to a parameterization suggested by Dall Amico et al. (2011). The soil composition is assumed to be constant, so that any changes in soil moisture other than those due to phase changes are ignored. This assumption is well justified as the soils at Samoylov are almost completely saturated (Langer et al., 2013). CG3 also includes a simplified snow cover representation that only takes into account a limited number of the natural processes that occur in snowpacks. It is therefore not comparable to more sophisticated snow models such as SNOWPACK or CROCUS. Therefore, in our simulations with CG3, the snow properties involved in conductive heat transfer were taken either from SNOWPACK simulations (in Sect. 5) or derived from an external construction (in Sect. 6), by-passing the CG3 estimates for these properties. All other properties or processes were calculated by CG3: this includes an exponential damping of incoming short wave radiation with snow depth, assuming a constant light extinction coefficient (e.g. O'Neill and Gray, 1972), and a snow albedo decreasing with snow ageing (Westermann et al., 2016).

**3 Thermal properties and spatial variability of the Samoylov snowpack in April 2013**

**3.1 Observations**

The stratigraphy, density and $K_{eff\text{-}z}$ profiles at the CT grass-center, ice-center, slope and rim sites are shown in Figure 3. The general characteristics of the CT profiles (grain types, snow depth, DH thickness-to-total snow depth ratio) for each micro-topographic class were similar to the median characteristics from the manual profiles for that particular micro-topographic class (Figure 4). The only exception was the CT slope profile, which featured an exceptionally high proportion of DH (80 %, while the median for slope sites was 50 %). Apart from that one exception, our four CT profiles were representative for their classes.





As in other tundra snowpacks described in previous publications, the Samoylov snowpack is largely made up of basal DH and of wind slabs with small rounded grains (RG). These layers exhibit significantly distinct densities and $K_{eff\text{-}z}$ values (Figure 5, p-values < 0.05 for a 2-sided t-test): the DH layers have a mean density of 236 kg m$^{-3}$ and a mean $K_{eff\text{-}z}$ of 0.22 W m$^{-1}$ K$^{-1}$, while wind slabs have a mean density of 356 kg m$^{-3}$ and a mean $K_{eff\text{-}z}$ of 0.36 W m$^{-1}$ K$^{-1}$. Some DH layers in the

slope, rim, and ice-center CT profiles have densities reaching up to 300 kg m$^{-3}$. These layers have probably been formed by the metamorphism of former wind-crusts (indurated DH), thereby retaining a high density. Similar layers were found in the manual profiles, exhibiting higher hand hardness (2 to 3) and smaller grain sizes (1 to 2 mm) than basal DH layers (hand hardness 1, grain size 5 to 10 mm). As in the CT profiles, these layers were found more than 7 cm above the vegetation layer, where wind effects are likely to be more pronounced.

Micro-topography and surface conditions clearly play a role in shaping the snowpack conditions. We found the snow to be significantly deeper at slope sites and shallower at rim sites (27 cm *vs*. 10 cm median depths) than at the center sites (p-value < 0.1 for a two-sided t-test). The rim sites are the most exposed to wind and receive reduced deposition during blowing snow events, while slopes, especially those on the lee side, experience lower wind speeds and enhanced deposition. The larger number of distinct snow layers in slope profiles is a further evidence of that process. These observations are generally

supported by those of Wainwright et al. (2017), who described significant negative correlations between surface elevation and snow depth at four different polygonal tundra sites. In contrast to snow depth, DH thickness-to-total snow depth ratios were lower on slopes and higher on rims (0.5 vs. 0.8 median ratios), although these differences do not appear to be significant. Rim profiles also exhibit a large proportion of DH-chains (Figure 4), i.e. vertically structured DH crystals in which most of the lateral bonds have disappeared (Fierz et al., 2009). This is in line with an increased temperature gradient at

rim sites as a result of shallower snow depths, which promotes active temperature-gradient metamorphism. Profiles from grass-center sites have characteristics in-between those of slope sites and rim sites, with 19.5 cm median snow depth and 56 % DH. Ice-center sites have similar snow depths to grass-center sites but a significantly lower proportion of DH than the other classes. Ice is more conductive than frozen ground, even if saturated. When ice is present at the bottom of the snowpack (due, for example, to the freezing of a pond) the temperature gradient within the snow is therefore reduced,

restricting DH growth. Basal DH crystals formed over ice are therefore smaller (4 mm to 6 mm) than those found in grass-center profiles (6 mm to 8 mm).

### 3.2 Spatial variability in bulk thermal properties

We calculated the bulk $K_{eff\text{-}z}$ ($K_{bulk}$) for each CT site by taking the $K_{eff\text{-}z}$ and the thickness of individual snow layers, and calculating the equivalent conductance of resistances in series. The $K_{bulk}$ value showed little variation between our three CT

sites with underlying grasses: $K_{bulk}$ was 0.21 W m$^{-1}$ K$^{-1}$ at the CT rim and slope sites and 0.23 W m$^{-1}$ K$^{-1}$ at the CT grass-center site (Figure 3). A more representative slope site with a lower proportion of DH portion would probably have had a slightly higher $K_{bulk}$ value. A much higher $K_{bulk}$ value was obtained in the presence of basal ice, where the development of DH is reduced: the value of $K_{bulk}$ at the ice-center site was 0.33 W m$^{-1}$ K$^{-1}$. We tested the assumption that differences in the



DH-thickness to total snow depth ratio (hereafter $\alpha$) can mostly explain the variability in Kbulk across the four CT sites. For this we relied on the approach used by Zhang et al. (1996), who considered that an Arctic snowpack can be approximated by two homogeneous layers, a DH layer and a wind-slab, each with its own distinctive density and $K_{eff\text{-}z}$ value. Rutter et al. (2014) also used a similar approach for microwave emission modelling. Following this approach, $K_{bulk}$ is expressed by:

$$K_{bulk} = \cfrac{1}{\cfrac{\alpha}{K_{DH}} + \cfrac{1-\alpha}{K_{crust}}} \tag{1}$$

where $K_{DH}$ and $K_{crust}$ are the $K_{eff\text{-}z}$ for DH and wind crust layers, which we approach by their mean values in our CT samples (0.22 W m$^{-1}$ K$^{-1}$ and 0.36 W m$^{-1}$ K$^{-1}$, respectively). $K_{bulk}$ is thus a decreasing function of $\alpha$. We found that 72 % of the variability in $K_{bulk}$ between our four sites can be explained by this simple 2-layer approach.

The insulating power of a snowpack is characterized by $R_{th} = HS / K_{bulk}$ (see Introduction). The variations in snow depth between our four sites, shaped by the micro-topography (see Section 3.1), modulate the variations in $K_{bulk}$ to produce quite

contrasting $R_{th}$ values. The ice-center profile has a very low $R_{th}$ (0.48 m$^2$ K W$^{-1}$) due to a high $K_{bulk}$ and a moderate snow depth. The $R_{th}$ value however increases from the rim site (0.57 m$^2$ K W$^{-1}$), through the grass-center site (0.87 m$^2$ K W$^{-1}$), to the slope site (1.59 m$^2$ K W$^{-1}$): this increase follows the increase in snow depth between these sites (from 10 cm to 19.5 cm and 27 cm, respectively), despite variations in the $K_{bulk}$ values (which at times also increase with snow depth). The CT grass-center site, for example, has both a greater snow depth and a higher $K_{bulk}$ value than the CT rim site, so that the net effect on

$R_{th}$ remains qualitatively equivocal.

Our observations suggest that, when there is basal vegetation present, $R_{th}$ is more sensitive to variations in total snow depth than to variations in the DH proportion $\alpha$, which controls $K_{bulk}$. We assessed this by looking at the sensitivity of $R_{th}$ to $\alpha$ and $HS$ in the 2-layer approach. $R_{th}$ is expressed by:

$$R_{th} = \frac{\alpha.HS}{K_{DH}} + \frac{(1-\alpha).HS}{K_{crust}} \tag{2}$$

implying a sensitivity to variations in $HS$ ($\frac{\partial R_{th}}{\partial HS}$) and a sensitivity to variations in $\alpha$ ($\frac{\partial R_{th}}{\partial \alpha}$) expressed by:

$$\frac{\partial R_{th}}{\partial HS} = \frac{\alpha}{K_{DH}} + \frac{1-\alpha}{K_{crust}} \tag{3}$$

$$\frac{\partial R_{th}}{\partial \alpha} = HS.\left(\frac{1}{K_{DH}} - \frac{1}{K_{crust}}\right) \tag{4}.$$

We estimated bounds of 3.5–4.3 m K W$^{-1}$ and 0.17–0.71 m$^2$ K W$^{-1}$ for these sensitivities, respectively, considering $\alpha = 0.4$–

0.9 and $HS = 0.1$–0.4 m. The $HS$ decreased by 0.1 m from the CT grass-center profile to the CT rim profile, while $\alpha$ increased by 0.22. From the median grass center profile to the median slope profile, $HS$ increased by 0.08 m while $\alpha$ decreased by 0.06. With these orders of magnitudes, it appears clearly that variations in $HS$ have a greater influence than variations in $\alpha$ on the insulating power of snow across the polygonal micro-topography when there is basal vegetation present.





### 3.3 Comparison of $K_{eff\text{-}z}$ observations with existing parameterizations

In the four CT profiles $K_{eff\text{-}z}$ showed a strong correlation with density (r = 0.94). We investigated the ability of three different parameterizations for $K_{eff}$ or $K_{eff\text{-}z}$ to match the values obtained with our measurements (Figure 6). These parameterizations are from Calonne et al. (2011), Riche and Schneebeli (2013) and Löwe et al. (2013), and we refer to them hereafter as

C2011, R2013 and L2013 respectively. C2011 expresses the mean of the vertical and horizontal components of $K_{eff}$ as a density-based regression. R2013 expresses the vertical component of $K_{eff}$ ($K_{eff\text{-}z}$) as a density-based regression inferred only from DH and faceted crystal (FC) samples that exhibit a vertical anisotropy. Finally, L2013 is a regression of $K_{eff\text{-}z}$ based on density and anisotropy. It relies on an anisotropy parameter, Q, calculated directly from CT images. Q is above 0.33 for vertically anisotropic samples and below 0.33 for horizontally anisotropic samples.

With respect to our data, there is an improvement in performance from C2011 (good correlation but noticeable bias) to R2013 (good correlation, reduced bias), and finally to L2013 (improved correlation and reduced bias). C2011 does not take anisotropy into account, nor does it attempt to represent the vertical component of the conductivity ($K_{eff\text{-}z}$), which probably explains its relatively poor performance. A bias in R2013 for snow types with horizontal anisotropy (Q < 0.33) is to be expected as R2013 is designed to represent the $K_{eff\text{-}z}$ of vertically anisotropic grains. Our results confirm that R2013 is indeed

biased on samples with Q < 0.33 (Figure 6b), consisting of RG and partly decomposed/fragmented particles (DF). R2013 also underestimates $K_{eff\text{-}z}$ in the samples with the greatest vertical anisotropy, which may be due to the very small number of samples (only 2) used by the authors to constrain their parameterization at densities greater than 300 kg m$^{-3}$. Being derived from a density-based regression, R2013 is furthermore structurally incapable of taking into account all possible degrees of anisotropy encountered in nature. The best performance was obtained with L2013, which confirms the importance of

anisotropy in $K_{eff\text{-}z}$ estimations. The two largest biases obtained from regressions based on density only (underestimations of $K_{eff\text{-}z}$ by 47 % and 49 %) were obtained using C2011 on DH-chains, i.e. on highly anisotropic grain forms.

### 4 Adaptations of the SNOWPACK model to the Arctic context at Samoylov

In the Introduction we recalled that adaptations were required to the current generation of snow models if realistic density profiles (and consequently $K_{eff\text{-}z}$ profiles) were to be simulated in Arctic conditions. These adaptations concerned wind

densification (WIND), the water vapour transport occurring under steep temperature gradients (VAP), and the mechanical, optical and metamorphic effects of basal vegetation protruding into the snowpack (VEG). The traditional density-based formulations for $K_{eff\text{-}z}$ also needed to improve and incorporate the effect of grain anisotropy (ANISO).

Some of the effects of VEG (mechanically reduced compaction, enhanced grain growth) and VAP (reduced density in the basal layers as a result of upward flux, enhanced grain growth) are hard to disentangle in Arctic conditions, where they both

contribute to density reduction and enhanced grain growth in basal layers. Furthermore, no explicit description of water vapour transport and associated metamorphism is available in the current snow models. We therefore chose to address both VAP and VEG together under a phenomenological "VEG" adaptation.



For the mechanical effect of VEG we reduced the fresh snow density ($\rho_0$) for snow that occurs within the grasses, i.e. up to a thickness of 7 cm. The underlying hypothesis is that grasses form a rigid structure that protects snow from wind compaction and introduces macroscopic voids that reduce its density. Different $\rho_0$ values were tested and 150 kg m$^{-3}$ was chosen as giving the best match to end-of-season in situ density observations. Dominé et al. (2016a) chose to increase the dry snow

viscosity in the CROCUS snow model by a factor of between 10 and 100, in order to take into account the limited snow compaction within the stems of shrubby vegetation. In our case, however, an alternative approach was required since self-compaction is very limited in the thin Samoylov snowpack.

The optical effect of VEG (i.e. the absorption of solar radiation by grasses and sandy impurities, which are common at Samoylov) was not taken into consideration but is addressed later in the Discussion section.

The metamorphic effect of VEG was addressed by enhancing bond and grain growth rates by a constant factor within the grasses-and-snow layer. This phenomenologically represents the favourable conditions for grain growth within airy vegetation layers. We feel justified in taking this approach because the current metamorphism and diffusion laws of the snow models are unable to reproduce the commonly observed grain sizes in excess of 10 mm in basal DH layers accommodating vegetation. A factor of 5 was selected as best reproducing the observed end-of-season DH grain sizes at Samoylov. Both

bond and grain growth rates were enhanced by the same factor in order to keep their ratio constant, as this ratio governs a number of mechanical and thermal properties in SNOWPACK.

For WIND, we built on the work by Groot-Zwaaftink et al., (2013) who designed an adaptation of SNOWPACK to Antarctica Dome C conditions. These authors considered that effective snow deposition on the surface occurs only during wind-events, i.e. periods when the wind speed averaged over 100 hours ($U_{100-h}$) exceeds a 4 m s$^{-1}$ threshold ($U_0$= 4 m s$^{-1}$).

The density of fresh snow ($\rho_{newsnow}$) is then a logarithmic function of $U_{100-h}$:

$$\rho_{newsnow} = \rho_0 + \Delta\rho . log \left( \frac{U_{100-h}}{U_0} \right) \tag{5}$$

The use of this approach is justified at Samoylov as wind conditions at the Samoylov station (mean annual wind speed 3.6 m s$^{-1}$) are comparable to those at Dome-C (mean annual wind speed 2.9 m s$^{-1}$), and more than 50 % of snow deposition at Samoylov occurs during wind events. Groot-Zwaaftink et al. (2013) used $\rho_0$ = 250 kg m$^{-3}$ as the lowest fresh-snow density.

However, no value as low as that was recorded during the 2013 program from the wind slab layers at Samoylov, where the density is always above 305 kg m$^{-3}$. Such densities are furthermore essentially achieved by wind compaction (settlement in thin arctic snowpacks is negligible). We therefore used $\rho_0$ = 305 kg m$^{-3}$ in Eq. (5). The original value for $\Delta\rho$ ($\Delta\rho$ = 361 kg m$^{-3}$) was retained.

For the ANISO adaptation we implemented in SNOWPACK a formulation derived from L2013 (Löwe et al., (2013), their

Eq. (5)), which by considering anisotropy, explained a larger part of the observed variability in our $K_{eff-z}$ measurements than formulations relying solely on density. However, L2013 requires an anisotropy parameter Q, which can either be calculated from CT images of samples, or estimated from polarimetric radar data (Leinss et al., 2016), but is not yet included in current snow models. In order to implement L2013 in SNOWPACK we therefore had to derive an empirical relationship between Q



and a modelled microstructural parameter. To this end, we used the data from Löwe et al. (2013) to obtain statistical regressions between Q and the optical equivalent diameter of snow grains. We calculated these regressions for different grain-type classes: rounded grains (RG), depth hoar (DH), faceted crystals (FC), decomposed-fragmented particles (DF), and melt forms (MF), most of which indicating reasonable linear dependences. These regressions were used in SNOWPACK in order to derive the parameter Q, using normalized grain size (within each grain type class) as a proxy for normalized optical diameter. We only took into account anisotropy for the RG, DH and FC grain types, as these are the dominant grain types in the Samoylov snowpack. Regressions coefficients and implementation details are given in Appendix A.

The three adaptations (WIND, VEG, and ANISO) can also be combined. Simulations were initially carried out for the default SNOWPACK setup (DEFAULT) and for each of these adaptations individually, but both the WIND and VEG adaptations proved to be essential for the Samoylov snowpack conditions to be reasonably well reproduced. Results are therefore shown in this paper for the following *setups*, each combining one or more *adaptations*:

- **DEFAULT**
- **WIND**
- **WIND + VEG**
- **WIND + VEG + ANISO**

## 5 Simulations of snow properties and ground thermal regime (grass-center site)

We carried out simulations with SNOWPACK and CG3 to represent the snow and ground conditions in the grass-center of the reference polygon, where the SNOWPACK snow forcing data were acquired (see Sect. 2.2.1) and CG3 soil properties calibrated (see Sect. 2).

### 5.1 Snow simulations

The adaptations to SNOWPACK enable a reasonable simulation of the Samoylov snowpack (Fig. 7 and Fig. 8), but both VEG and WIND adaptations are critical. While all setups consistently produce a thick basal depth hoar layer at the end of the season, DEFAULT simulates a density profile that has too low a mean value (190 kg m$^{-3}$) when compared to the CT grass-center (290 kg m$^{-3}$) and to the average value for the four CT profiles (279 ± 34 kg m$^{-3}$). This simulated density profile is also inverted, featuring higher values at the bottom and illustrating the typical bias highlighted by Dominé et al. (2016b) and Barrere et al. (2017). Bulk $K_{eff-z}$ obtained using DEFAULT is likewise too low compared to observations (0.11 *vs*. 0.23 W m$^{-1}$ K$^{-1}$ for the CT grass-center), and is also inverted. This low bias is likely to have caused the rapid growth of DH in this setup, as a low $K_{eff-z}$ favours steep temperature gradients. The low density and $K_{eff-z}$ biases can be corrected by using the WIND option, which in its current form tends to overestimate bulk density. However, the WIND option alone produces quite flat (i.e. vertically uniform) density and $K_{eff-z}$ profiles. The VEG adaptation is then needed to produce a correct shape for these profiles, with higher values at the top and lower values at the base. Thus while the WIND option on its own reduces the





DH growth due to dense and conductive bottom snow, the addition of the VEG option introduces lower densities and $K_{eff\text{-}z}$ values for the basal layers and permits a more rapid and thicker growth of DH.

Combining the WIND and VEG options therefore yields reasonable simulations of bulk $K_{eff\text{-}z}$ (0.20 W m$^{-1}$ K$^{-1}$) and density (305 kg m$^{-3}$). When the ANISO option is introduced (WIND+VEG+ANISO), the simulated bulk $K_{eff\text{-}z}$ (0.24 W m$^{-1}$ K$^{-1}$) also

agrees well with the CT grass-center estimate (0.23 W m$^{-1}$ K$^{-1}$), while the inter-layer variability in $K_{eff\text{-}z}$ is enhanced, thus better reflecting the observed inter-layer variability (Figure 8). It is interesting to note that both the WIND+VEG and the WIND+VEG+ANISO setups produce a DH layer that is up to 10 cm thick at the end of the snow-season, above the vegetation layer: this means that former wind-crusts have been transformed into DH, producing the indurated DH layers reported in observations.

Finally, all SNOWPACK setups produce a thick layer of faceted crystals in the upper part of the snowpack, but faceted crystals were rare in the late April 2013 Samoylov snowpack (Figure 4). We interpret this as a likely bias in SNOWPACK that results in too rapid formation of faceted crystals. On the other side it is possible that a wind event on 10 April 2013 contributed to the high amount of RG found in the April 21 manual and CT profiles. Because it brought a very low accumulation at the SR50, this event was not captured in simulations with the WIND option.

**5.2 Soil simulations**

The ground thermal regime at the grass-center of the reference polygon was simulated by CG3 over the 2012-2013 snow season using snow properties calculated in SNOWPACK with the DEFAULT, WIND, WIND+VEG and WIND+VEG+ANISO setups, respectively. These simulations were compared with the soil temperature measurements from the same grass-center site. The reference polygon also hosts soil temperature measurements from a rim and a slope site: the

spatial variability reflected in these three measurements was also considered and is referred to as "observed variability" in soil temperatures in both text and figures.

To analyse the modelling performances we split the winter into 4 phases:

- **Phase 1 – freezing**: 1 October (snow onset) to 7 November
- **Phase 2 – cooling**: 7 November to 20 February (dark winter followed by a period with low-angle solar radiation)
- **Phase 3 – warming:** 20 February to 5 May (melt-out date)
- **Phase 4 – thawing**: 5 May to 31 May

The WIND, WIND+VEG, and WIND+VEG+ANISO setups produced soil temperatures in good agreement with the grass-center measurements (Figure 9, Table 1), especially during freezing and cooling phases: the deviation from the measured soil temperatures when using the WIND+VEG+ANISO setup was of the same order of magnitude as the observed variability,

while the deviations when using the WIND and WIND+VEG setups were slightly greater. The DEFAULT setup yielded a clear overestimation of soil temperatures at all depths, which could not be explained by the observed spatial variability in soil temperatures. This bias started during the freezing phase and persisted throughout the snow season; it is likely to be caused by the underestimation of $K_{eff\text{-}z}$ in the DEFAULT setup (see Sect. 5.1), which also starts in the early snow season




during rapid DH formation. In light of the good agreement between our $K_{eff\text{-}z}$ estimates by CT and the simulated $K_{eff\text{-}z}$ profiles in the WIND+VEG+ANISO setup (Sect. 5.1), we interpreted these results as confirming the soundness of our CT estimates for $K_{eff\text{-}z}$.

The performance of the WIND, WIND+VEG and WIND+VEG+ANISO setups deteriorated during the course of the warming phase, when all simulations showed at first a systematic warm bias, which then turned into a cold bias at the start of the thawing phase. The warm bias during the warming phase suggested that limitations exist in the modelling of energy transfer processes within the snow, as here modelled by CG3. We formulated two hypotheses:

- Deficiencies in the parameterization of radiative heating within the snowpack may be involved as the bias concurs with the increase in shortwave radiation.

- The formation of an air layer at the base of the natural snowpack (as a result of mass depletion due to a sustained upward vapour flux throughout the winter) may increase its insulating power as the season advances. The formation of such an air layer within an Arctic context has previously been reported by Dominé et al. (2016b) but is not represented the adapted SNOWPACK and therefore in the thermal properties passed to CG3.

We tested the thermal impact of both hypotheses by conducting sensitivity simulations in which:

(i)    The penetration of radiation into the snowpack was switched off in the CG3 model. This was done for the four SNOWPACK setups.

(ii)    We inserted an air-layer (with $K_{eff\text{-}z} = 0.024$ W m$^{-1}$ K$^{-1}$) at the base of the snowpack during the warming phase, growing in a linear fashion from 0 to 1.5 cm during the warming phase. This was done by modifying the snow properties from the WIND+VEG+ANISO setup, and resulted in a linear reduction in bulk $K_{eff\text{-}z}$ from 0.23 to 0.16 W m$^{-1}$ K$^{-1}$ over that period.

Suppressing the penetration of solar radiation in the snowpack considerably reduced the warm biases in soil temperatures during the warming phase for all WIND setups, while leaving their performances during the freezing and cooling phases unaffected (Figure 10). While physical reasons for a likely bias in radiative transfer in CG3 will be advanced in section 7, the remaining simulations in this study were carried out with the solar radiation penetration switched off. The air-layer hypothesis did not, however, lead to any visually identifiable change in the simulations. This reveals a very low sensitivity of the soil thermal regime to variations in snow thermal conductivity during the warming phase.

## 6 Thermal implications of snow spatial variability

The observed variability in soil temperatures at rim, slope and grass-center sites probably only captures part of the thermal impact of the snow spatial variability at Samoylov. This may be firstly because of the small sample size (only one rim, one slope, and one grass-center soil temperature measurement site); secondly because the snow properties on top of the soil sensors were not measured so that the representativity of snow conditions at the soil sites with respect to their micro-





topographic class is unknown; and thirdly, because the soil temperature observations are also affected by spatial variability in the soil's thermal properties, which may interfere with any thermal effect solely due to snow variability.

To more thoroughly assess the thermal impact of snow spatial variability across the polygonal tundra at Samoylov, we made use of the transect data from the reference polygon, which allowed to retrieve a large sample of snow conditions, and we

combined different steps to estimate relevant time-series of snow properties along the transect and carry out corresponding soil temperature simulations. Note that we excluded ice-center conditions from this assessment. More precisely, we proceeded as follows: DH thicknesses and snow depths were extracted from 31 points at a 50 cm spacing along the transect, by manual post-processing of the NIR images (Fig. 1e). Thus the DH-thickness to total snow depth ratio $\alpha$ could be calculated at each point. The 2-layer approach by Zhang et al. (1996), which was described briefly in Sect. 3, was then used

to infer bulk $K_{eff\text{-}z}$, $R_{th}$, and density values for the snow at these 31 points, relying on Eq. (1) and Eq. (2). Time series of the snow properties were however needed to simulate soil temperatures at the 31 points over the entire 2012-2013 winter season. We therefore constructed these time series using the 2-layer approach and estimated time series for the evolution of snow depth and DH thickness, and hence for $\alpha$, constructed in such a way as to match the $\alpha$ from the NIR images at the end of the snow season (April 2013). The hypotheses behind the construction of these time-series, together with other relevant details,

can be found in Appendix B. The wind-slab density and $K_{eff\text{-}z}$ were considered to remain constant over time and to be equal to the means of values measured in these layers at the CT sites in April 2013 (Table 2). The DH density and $K_{eff\text{-}z}$ varied over time according to the WIND+VEG+ANISO simulation; their values on 21 April 2013 are reported in Table 2. CG3 simulations were then performed with these time series for each of the 31 transect points.

As with the results from the four CT profiles (Sect. 3.2), there is a moderate spread in the simulated $K_{eff\text{-}z}$ ensemble (from

0.22 W m$^{-1}$ K$^{-1}$ to 0.29 W m$^{-1}$ K$^{-1}$ over the entire winter), while the spread of $R_{th}$ values is much greater (from 0.45 m$^2$ K W$^{-1}$ to 1.2 m$^2$ K W$^{-1}$) and reaches a maximum during the warming period (Fig. B2). Note that our CT profiles effectively captured this spread of $R_{th}$, values, with 0.48 m$^2$ K W$^{-1}$ for the CT rim profile and 1.59 m$^2$ K W$^{-1}$ for the CT slope profile at the end of the warming period. The latter value actually exceeds the $R_{th}$ envelope from transect data as a result of the exceptionally high proportion of DH in the CT slope profile.

The spatial variability in snow insulation results in a pronounced spread of the simulated soil temperatures, which we refer to as "modelled variability" in the relevant figures (Figure 11). This modelled variability encompasses the spatial variability in soil temperatures represented in the rim, slope and center measurements (observed variability, Figure 11), which is a desirable feature. However, the simulated spread is much greater than the spread in the observations, especially during the cooling phase when it reaches 6.3 °C at 5 cm depth while the observed spread does not exceed 2 °C. The reasons advanced

earlier in this section to justify a complementary modelling approach for spatial variability assessment probably explain part of this difference. In addition, the measured rim and slope temperatures, which determine the maximum amplitude of the spread in the observations, responded differently at the beginning of the cooling phase, with the temperature dropping rapidly for the slope profile in early November but only gradually for the rim profile. This behaviour reversed from early December until the end of the cooling phase, with the spread in observed temperatures between a colder rim and a warmer



slope reaching its maximum. The contrasting behaviour of rim and slope in November probably limited the observed spread. Contrasting early-season wind erosion/deposition between the slope and rim profiles, together with differences in the late autumn soil water content, may have affected the zero-curtain duration and soil cooling dynamics. Neither of these phenomena are captured by our simple modelling.

During the warming period the variabilities in both modelled and observed soil temperatures are considerably reduced. Warming from the air is more efficient at sites with little snow insulation, which have the coldest soil temperatures during the cooling phase, than at sites with more snow insulation. This explains the reduction in the spread of soil temperatures after the month of April. However, the reduction in the spread of simulated soil temperatures starts earlier, in late February. This again indicates a reduced sensitivity of the ground thermal regime to variations in the thermal properties of the overlying

snow during the whole warming phase (cf. the sensitivity experiment with the insertion of a basal air layer in Sect. 5.2). This reduced sensitivity will be analysed in section 7 below.

In section 5 we compared the soil temperatures simulated for the grass-center site to our actual measurements at rim, slope and grass-center sites. The DEFAULT snowpack setup was rejected as it yielded soil temperatures that were too far above the observed range. The approach followed in this section allows a complementary estimation of the spatial variability in soil

temperature induced by spatial variations in snow (both in thickness and thermal properties). Despite a spread in simulated soil temperatures larger than the observed range, our conclusions regarding the DEFAULT version of SNOWPACK remain unchanged as the DEFAULT soil temperatures are still beyond the range of the simulated spatial variability.

## 7 Discussion

### 7.1 Comparison with snow data from similar contexts

The Samoylov snowpack shows similarities in its stratigraphy with Arctic snowpacks described previously by Dominé et al. (2015, 2016b) and Derksen et al. (2009). The tundra snowpacks investigated by these authors along a sub-arctic traverse comprised on average 65 % DH and had a mean density of 319 kg m$^{-3}$. Both of these values are close to those from Samoylov (54 % and 279 kg m$^{-3}$, resp.). The minor differences are probably due to differences in the wind conditions and the specific micro-topography of Samoylov, where some samples were collected from wind-sheltered slope/center sites or over

frozen ponds. Derksen et al. (2009) also investigated the differences between snowpacks overlying lake ice, river ice, and tundra sites, identifying larger proportions of DH over ice, which is contrary to our own results. However, their study considered lake or river ice overlying liquid water that is warmer than the surrounding soil. This thermal contrast enhances the development of faceted grains. In contrast, the end-of-summer water level at the sampled ice-center site on Samoylov was shallow, and shortly after freezing the ice extended to the ground, so that there could not be any enhanced thermal

contrast created by an underlying, relatively warm, body of liquid water.

There are few published observations or reports on the thermal properties of Arctic tundra snow. To our knowledge, the Samoylov samples are among the first samples of tundra snow to be analysed by CT. Publications by Dominé et al. (2015,



2016a, 2016b) and Barrere et al. (2017), which relied on NP measurements and a refined retrieval algorithm for $K_{eff}$, probably provide the most extensive thermal characterization of Arctic and sub-arctic snowpacks in recent years. These authors reported values of $K_{eff}$ lower than our $K_{eff-z}$ estimates, both for DH layers and for the bulk snowpack. Barrere et al. (2017) measured $K_{eff}$ values no higher than 0.12 W m$^{-1}$ K$^{-1}$ for basal DH in the May 2014 and 2015 snowpacks at Bylot Island (Baffin Island, Canada); they however reported much higher conductivities (0.37 W m$^{-1}$ K$^{-1}$) for indurated DH. After correcting for a 20 % systematic error associated with the NP method, these authors calculated bulk $K_{eff}$ values of less than 0.1 W m$^{-1}$ K$^{-1}$ for the 2014 and 2015 Bylot snowpacks, resulting in highly insulating snow (bulk $R_{th}$ values of 2.6 and 5.8 m$^2$ K W$^{-1}$). We estimated a bulk $R_{th}$ of 0.87 m$^2$ K W$^{-1}$ for our CT grass-center profile and a high upper bound of 1.59 m$^2$ K W$^{-1}$ for the CT slope profile. The $R_{th}$ values obtained by Barrere et al. (2017) indicate insulation that is closer to the end-of-season insulation simulated by the DEFAULT setup in SNOWPACK ($R_{th}$ = 1.75 m$^2$ K W$^{-1}$ in April 2013). This setup led to an overestimation of February soil temperatures at Samoylov by about 6 °C. Such a bias can hardly be explained by the spatial variability in snow conditions (see Sect. 6). Despite the disagreement with published estimates for $K_{eff}$ under similar conditions, the consistency of the CT estimates for $K_{eff-z}$ with recent parameterizations and with measured soil temperatures after combined snow-soil modelling provides some confidence in them. The Samoylov snowpack appears more conductive than the 2013-2014 and 2014-2015 snowpacks observed at Bylot Island. Furthermore, our results compare very well with the conductivities obtained using inverse modelling by Jafarov et al. (2014) at Deadhorse (Alaska), a site with snow and meteorological conditions similar to Samoylov.

We estimate that the ground temperature spread induced solely by snow spatial variability can reach 6.3 °C in the coldest part of the winter at Samoylov (Sect. 6). This estimate is consistent with those in previous publications: Sturm and Holmgren (1994) observed maximum differences in ground surface temperatures of up to 19.1 °C and mean winter temperature differences of up to 7.2 °C, between the tops and hollows of grass tussocks at Imnavait Creek, Alaska. Their investigations focused on smaller scale micro-relief (tenths of a cm) than ours, resulting from grass tussocks in the tussock tundra. Our study complements the sensitivity study by Zhang et al. (1996), who found a 12.6 °C spread in winter ground surface temperatures following an increase in the proportion of DH from 0 % to 60 % at West Dock near Prudhoe Bay, Alaska. This study included neither an observation-based range of the proportions of DH in the snowpack, nor the effect of co-varying DH thickness and snow depth. Furthermore, the DH and wind crust properties were kept constant over time. More recently, Gisnas et al. (2016) found a variability in ground temperatures of up to 6 °C in the Norwegian mountains, as a result of spatial variations in snow depth.

## 7.2 Light penetration in the Samoylov snowpack

The penetration of solar radiation in the natural snowpack at Samoylov is likely to be reduced by wind-blown sediments within some of the snow layers (Boike et al., 2003) and by the dense wind crusts at the top of the snowpack (Libois, 2014). While absorption of solar light in these layers may result in a localized increase in temperature within the snowpack, it is unlikely to have much warming effect on the underlying snow and soil because of the insulating nature of the snow. Brun et



al. (2011) had to reduce the penetration depth of solar radiation in the CROCUS snow model in the same way that we did, in order to reproduce the snow temperatures at depths greater than 20 cm within the Antarctic snowpack at Dome C (Brun, E. personal communication). Libois, 2014 modelled a temperature reduction of ~ 7 °C at 20 cm depth in the Dome-C snowpack in summer as a result of spatial variations in density between 150 and 300 kg m$^{-3}$ and consequent reduction in the penetration depth of solar radiation. Although radiative transfer models exist with fine spectral resolution that are able to circumvent this bias (Libois et al., 2013; Libois, 2014), these complex schemes are not implemented by default in operational snow models, which tends to hinder a proper representation of the underlying snow and soil thermal regime.

### 7.3 Temporal variations of the soil thermal sensitivity to snow properties

A key result of our ensemble simulations and observations is the increase in spatial variability in soil temperatures during the winter cooling phase and its reduction during the warming phase (Figure 11). We ascribe this behaviour to two physical mechanisms. First, winter cooling is characterized by very steep temperature gradients between atmosphere and soil (about 150 K m$^{-1}$; see Fig. C1 in the Appendix), which are later reduced and eventually vanish during the course of the warming phase. From Fourier's law for vertical heat flux ($q$):

$$q = -K_{eff-z} \cdot \frac{\partial T}{\partial z} \tag{6}$$

it is apparent that the sensitivity of the heat flux to $K_{eff-z}$ is the temperature gradient. The greatest impact of spatial variations in $K_{eff-z}$ on ground temperatures is therefore expected to occur when temperature gradients are at a maximum (i.e. during the cooling phase), while a far smaller impact is expected when temperatures gradients are low (i.e. during the warming phase). Second, the reduction in the temperature gradient during the warming phase allows the soil temperatures to equilibrate laterally. At locations with more conductive snowpacks (e.g. polygon rims) the soil responds more rapidly to warming air, which further reduces the difference between these soil temperatures and those in more insulated locations (e.g. polygon slopes): this also contributes to the reduction in spatial variability of soil temperatures during the warming phase.

### 7.4 Limitations of our approach and perspectives

In Arctic snowpacks the water vapour flux induced by the steep temperature gradients redistributes ice mass from basal to upper snow layers, so that the density of the basal layers may actually decrease unless there is compensation through moisture flux from the soil. On the basis of Eq. (7) in Riche and Schneebeli (2013) and snow temperatures simulated with the WIND+VEG and WIND+VEG+ANISO options, we estimate that about 2 kg m$^{-2}$ of ice is redistributed at Samoylov by this process between October and March. Unless sustained by soil water this flux could lead to a 1.3 cm thick ice-depleted layer at the base of the snowpack (assuming a basal density of 150 kg m$^{-3}$). The magnitudes of soil and snow vapour fluxes are not currently well constrained by observations, and they are not represented in detailed snow models such as SNOWPACK or CROCUS. To bypass these shortcomings and still produce reasonable SNOWPACK simulations, we adopted a phenomenological parameterization for the combined effects of snow vapour flux and vegetation on basal snow



porosity. Neither this approach nor the current observational datasets allow the retrieval of any dynamics in basal snow ice-depletion. A considerable uncertainty therefore remains regarding the thermal properties of snow in the early winter (cooling) period, when the sensitivity of ground thermal regimes to snow conditions is at its maximum. This uncertainty also affects our estimates of the thermal impact of snow spatial variability. Continuous monitoring of ice depletion at the base of

the snowpack and snow monitoring programs focusing on the early and dark winter periods would help to provide better constraints for the thermal characteristics of the snowpack and the underlying metamorphic processes at this time, yielding substantial benefits for the next generation of coupled snow-soil models.

It also appears indispensable to include a more systematic and comprehensive treatment of anisotropy in snow models than the coarse diagnostic based on grain size and type that we have used, with a consistent link between water vapour flux,

temperature gradient metamorphism, and anisotropy and with feedbacks on the mechanical (Srivastava et al., 2016), thermal, and optical properties of the snow. A promising way to further assess the relevance of anisotropy to the conductivity and the ground thermal regime may be to incorporate remote sensing observations. It has been recently demonstrated (Leinss et al., 2016) that the depth-averaged anisotropy parameter (Q) of a snowpack can be estimated from polarimetric radar data such as, for example, that available from the TerraSAR-X satellite. Such an analysis could be used to produce global maps of the

average anisotropy of snowpacks, as an indication of their metamorphic state.

Our combined SNOWPACK and CG3 simulations show a cold bias during and after melt-out. Hydrological processes within the snowpack related to thaw and rain are known to have an important influence on soil thermal dynamics, as has been emphasized in a large number of publications (e.g. Marsh and Woo, 1984a, b; Putkonen, and Roe, 2003; Westermann, 2009). In naturally stratified snowpacks, water percolation and the associated heat transfer during early melt periods occur in part

through "flow fingers", which are preferential infiltration paths through the snow cover that penetrate into the colder substrata (snow layers or soil), where they refreeze, releasing latent heat (Marsh and Woo, 1984a, b). This process is known to delay the bulk melting of the snowpack, while at the same time accelerating soil warming. Progress has recently been made in the representation of preferential flow features by applying the Richards equation to water flow within a snow matrix (Wever et al., 2015; D'Amboise et al., 2017), but their impact on soil temperatures has not yet been assessed. Snow

schemes used in permafrost models such as CG3 do not currently represent these processes, inducing significant biases in the melt period.

Finally, we assessed the thermal impact of snow spatial variability on the ground thermal regime. Our method takes into account snow variability linked to micro-topography but disregards the variability in soil properties and soil saturation, which are also related to micro-topography. Distributed simulations that include the effects of wind redistribution and spatial

variations in soil conditions could, in theory, support a more consistent assessment of spatial variability in soil temperatures but would require a considerable amount of in situ data for both calibration (of, for example, soil properties) and validation (Kumar et al., 2016). Models that have lower degrees of complexity but inherently account for spatial variability in snow and soil conditions within a statistical framework (e.g. Gisnas et al., 2016) provide a promising alternative and will benefit from the enhanced understanding that we have achieved of the links between micro-topography and snow insulation.



## 8 Conclusion

Mixing in-situ observations, cold laboratory analysis, and modelling, our work contributed to an improved characterization and understanding of the properties and spatial variability of an Arctic polygonal tundra snowpack and its role in shaping the underlying permafrost thermal regime during winter. Snow depth, which showed a strong correlation with micro-
topographical features, was found to be a crucial driver of the insulating power of snow over vegetated surfaces. The proportion of DH in the snowpack, which showed a weaker correlation with micro-topography, introduced a second-order control. Water-logged polygon centers in which basal ice forms during winter, were an exception to this rule of thumb due to weak DH formation resulting in conductive snowpacks despite intermediate snow depths.

The CT technique allowed estimates to be made of the thermal conductivity and anisotropy of Arctic snow samples that were
mainly of depth hoar and wind slabs with rounded grains. The retrieved properties confirmed the validity of a recent anisotropy and density-based parameterization of $K_{eff\text{-}z}$, that had not previously been tested on Arctic snow samples. A comparison with other regressions for $K_{eff\text{-}z}$ highlighted the importance of taking anisotropy into account in $K_{eff\text{-}z}$ formulations, especially for depth hoar.

Phenomenological adaptations to the SNOWPACK snow model, related to wind densification and the combined effect of
basal vegetation and strong water vapour flux in the lower snowpack, enabled the simulation of snow density and $K_{eff\text{-}z}$ profiles in good agreement with our CT estimates. Introducing anisotropy considerations in the formulation of $K_{eff\text{-}z}$ used in the model resulted in further improvements. These adaptations jointly allowed improved simulations of the soil temperatures, providing further support for the soundness of our CT estimates for $K_{eff\text{-}z}$.

We also estimated the impact of the natural snowpack spatial variability on the underlying permafrost thermal regime during
an entire winter based on our $K_{eff\text{-}z}$ and density observations and on our understanding of the snowpack dynamics. Beyond this quantitative estimate, which is intrinsically tied to the local climatology and micro-topography of our site, an important conclusion is that the sensitivity of the ground thermal regime to the overlying snow reaches a maximum during the cooling winter period, when temperature gradients between atmosphere and soil are at their steepest. It is therefore crucial to better constrain the thermal properties of snow and the relevant processes during the first half of the winter, a period that is often
less well monitored due to the dark and harsh winter conditions.

Finally, our study pinpointed remaining shortcomings of current snowpack and snow-permafrost models, which may be of interest to permafrost and climate modellers.



## 9 Figures and Tables

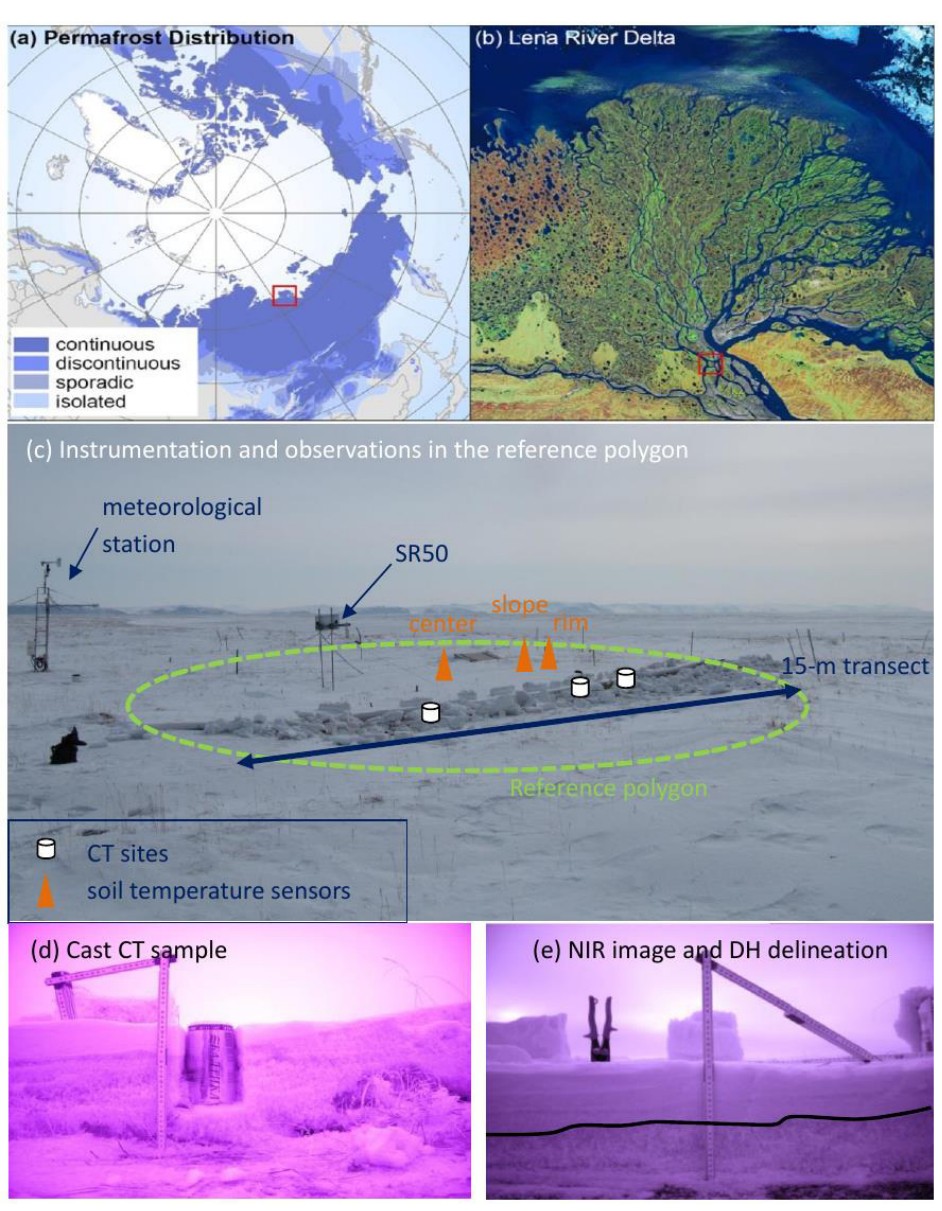

**Figure 1: Location of the Samoylov permafrost observatory within the continuous permafrost zone, Lena River Delta (a, b); instrumentation and observations in the reference polygon (c); cast CT sample (d); and NIR image of a transect's wall with the upper boundary of the DH layer delineated (e). See main text for abbreviations.**




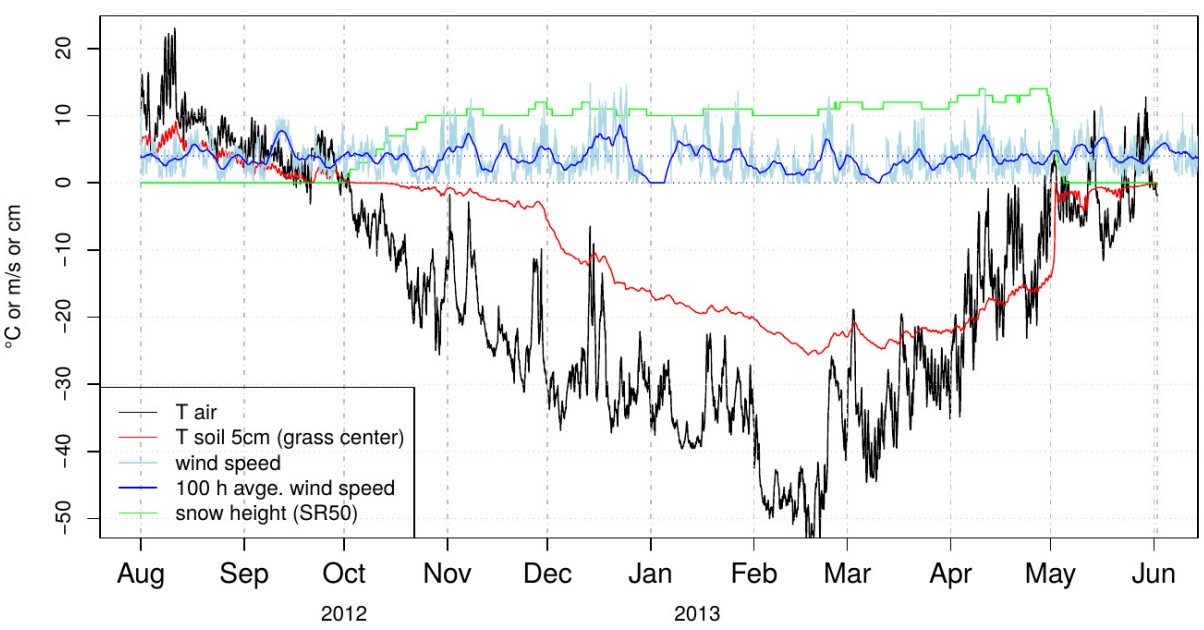

**Figure 2: Meteorological, snow and soil conditions at Samoylov over the 2012-2013 snow season.**



**Figure 3:** Stratigraphy (a) and density and $K_{eff-z}$ profiles (b) from the four CT sites. The abbreviations for the main grain types come from Fierz et al., 2009: PP=precipitation particles, DF=decomposing and fragmented precipitation particles, RG=rounded grains, FC=faceted crystals, DH=depth hoar, DHch=chains of DH, MF=melt forms. When several grain types coexist, the dominant type is listed first.





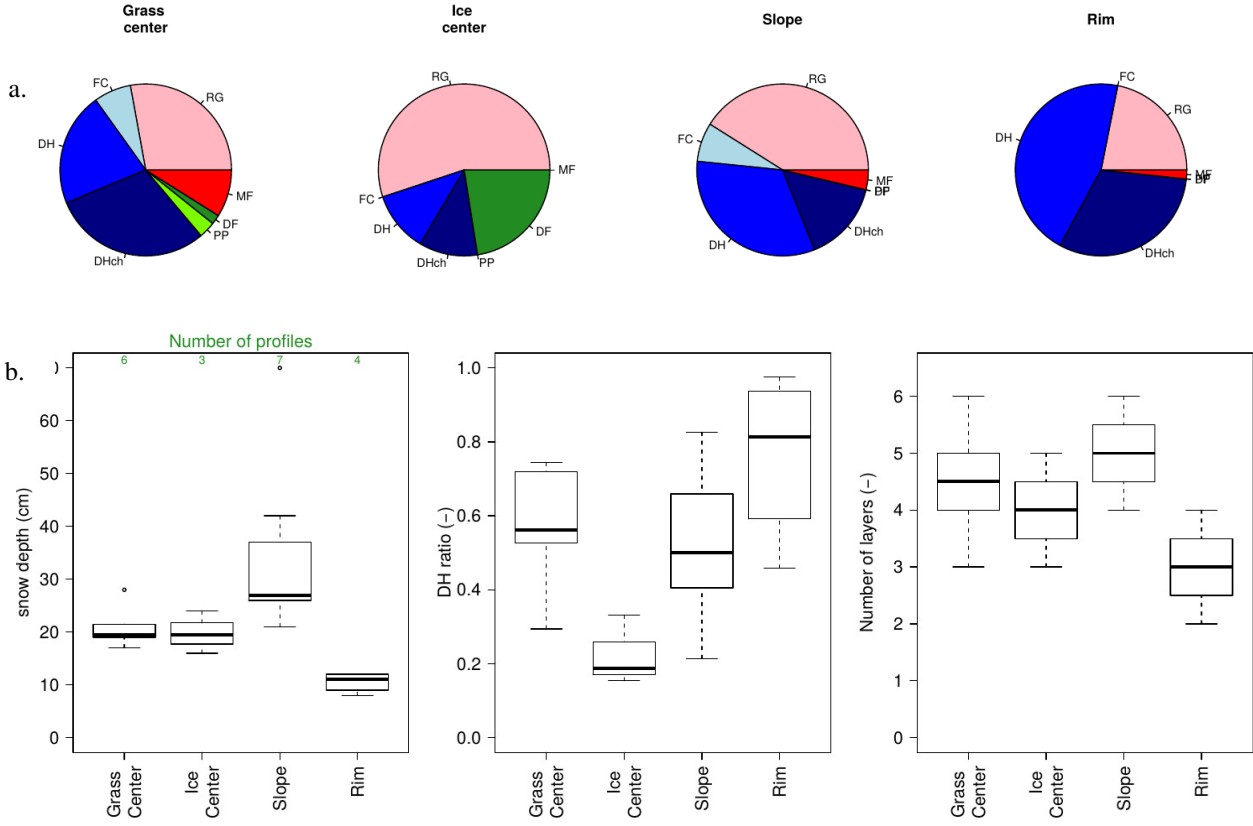

**Figure 4: Mean composition (a) and median characteristics (b) of the Samoylov snowpack in the four micro-topographic classes.**
These statistics include the observations from the 16 snowpits and the four CT sites. DH ratio is the DH thickness-to-total snow depth ratio, also called $\alpha$ in the manuscript.





a.

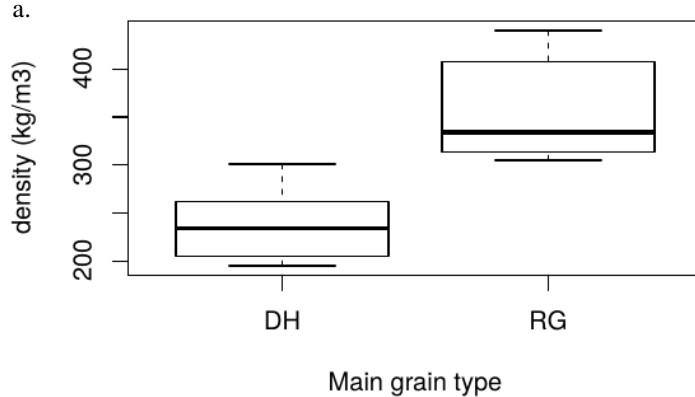

b.

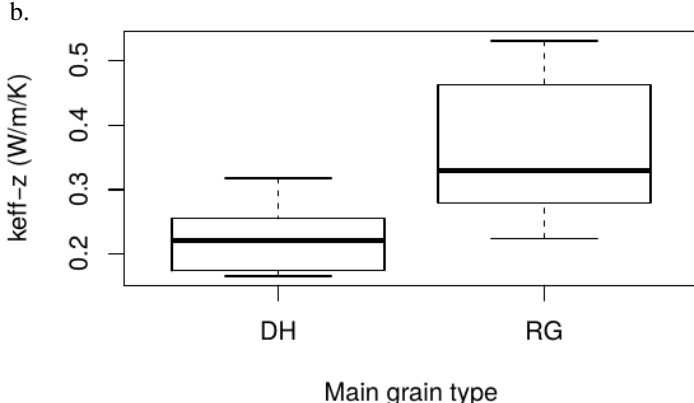

**Figure 5: Density (a) and $K_{eff\text{-}z}$ (b) for individual DH layers (11) and RG layers (8) at the CT sites. RG was the dominant type in the RG layers but could occasionally be associated with FC and DF. See Figure 3 caption for an explanation of the abbreviations.**





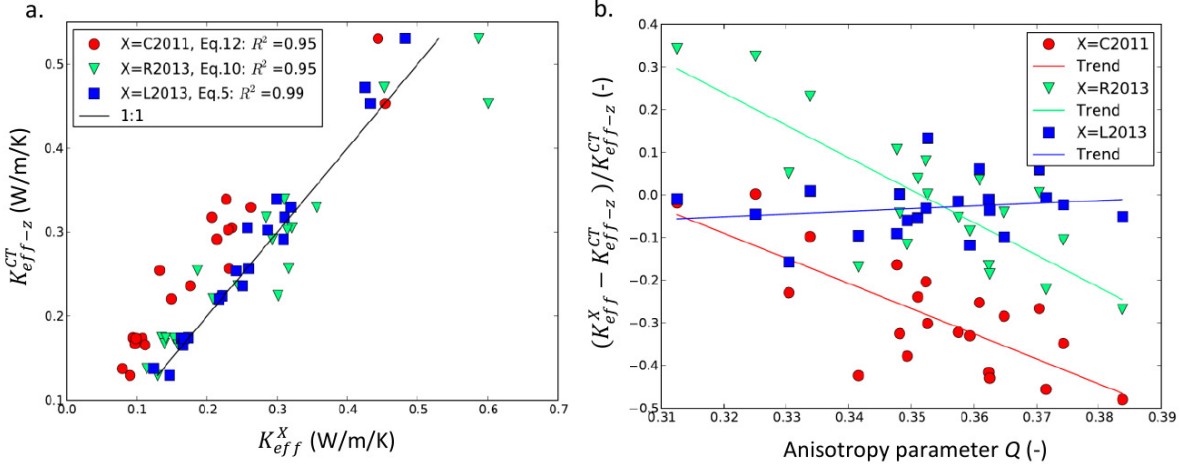

**Figure 6 : a. Comparison between estimates of $K_{eff}$ or $K_{eff-z}$ made with the CT method ($K_{eff-z}^{CT}$), and estimates made using parameterization "X" ($K_{eff}^{X}$, where X=C2011, R2013 or L2013: see manuscript for description of these parameterizations). b. Relative bias in $K_{eff}^{X}$ with respect to $K_{eff-z}^{CT}$ as a function of the anisotropy parameter Q. Each point represents a snow sample analysed by CT in this study.**



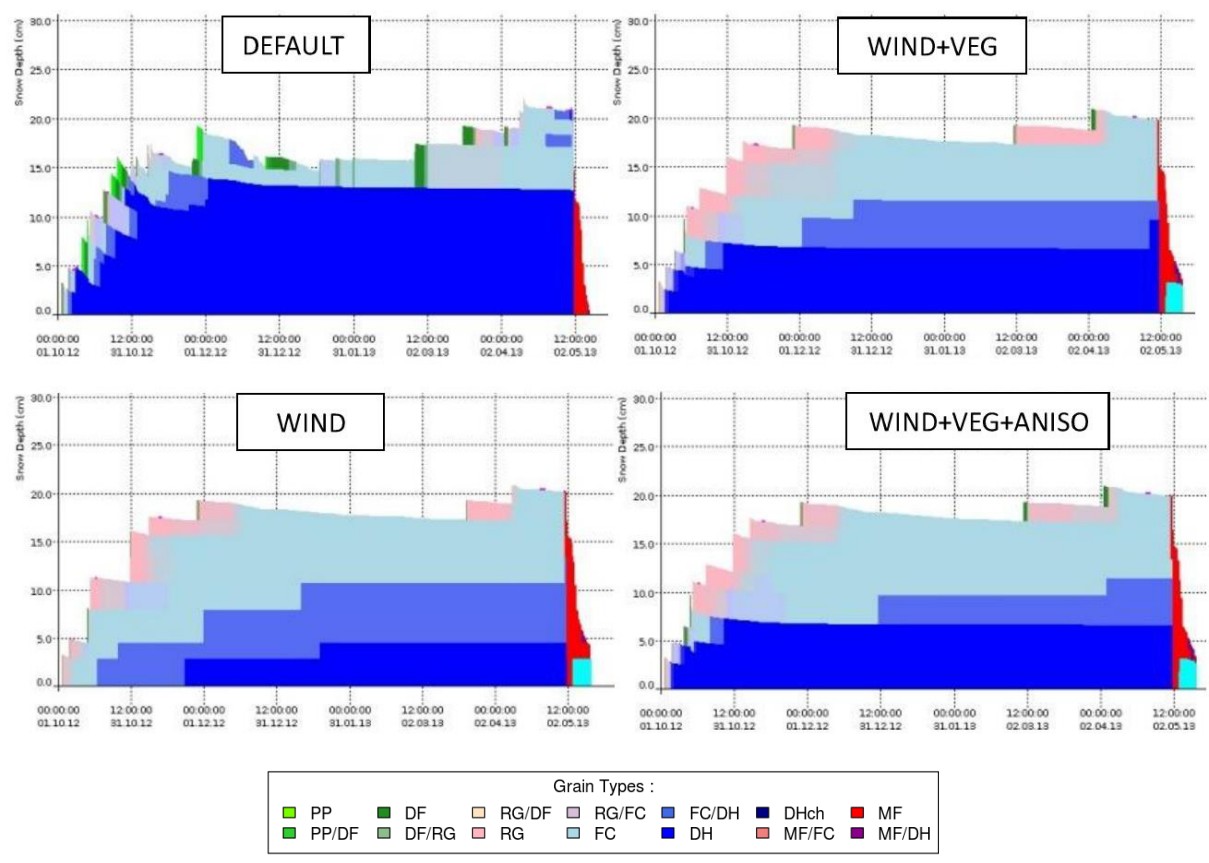

**Figure 7: SNOWPACK grain types in the 4 simulation setups.**



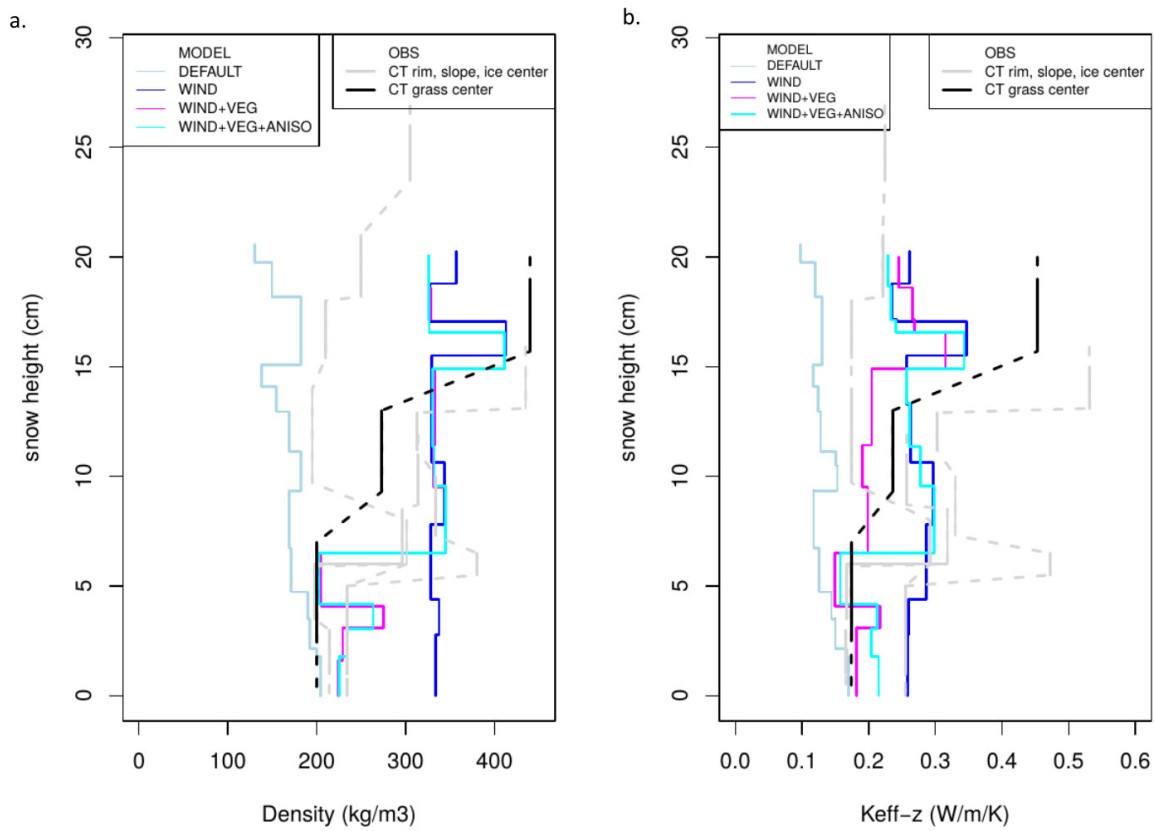

**Figure 8: Observed and simulated density and $K_{eff-z}$ profiles on 20-04-2013. Observations (OBS) are estimates made using the CT method at the four CT sites. Simulations (MODEL) were carried out with the four SNOWPACK setups.**







**Figure 9: Simulated vs. observed soil temperatures at depths of 5 cm, 20 cm, and 50 cm in the reference polygon's grass-center. OBS-variability (grey shading) is the envelope of observed soil temperatures from the monitored rim, center, and slope soil sites.**



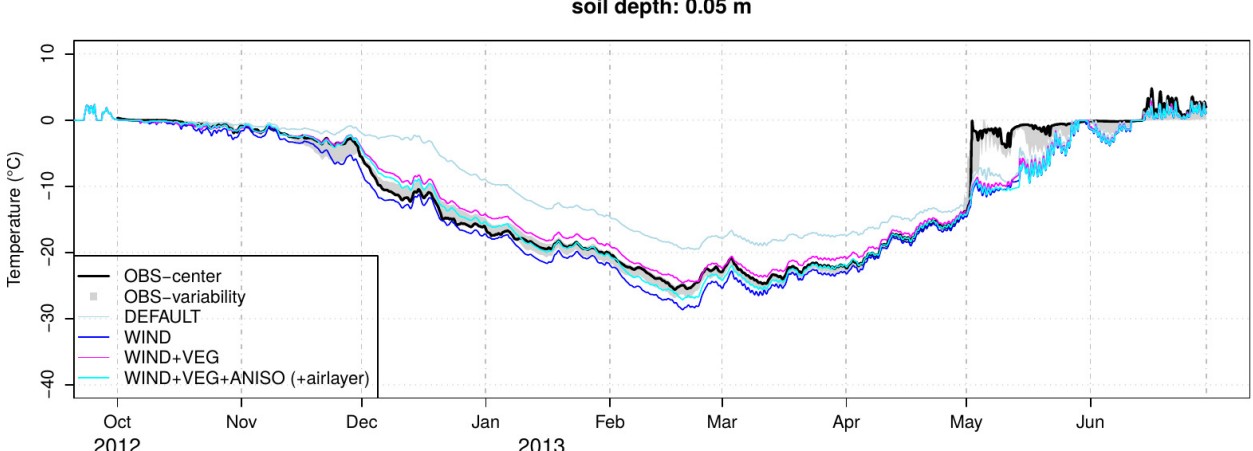

**Figure 10: As for Fig. 9 but with radiative transfer in snow switched off and the air-layer scenario added to the WIND+VEG+ANISO option.**

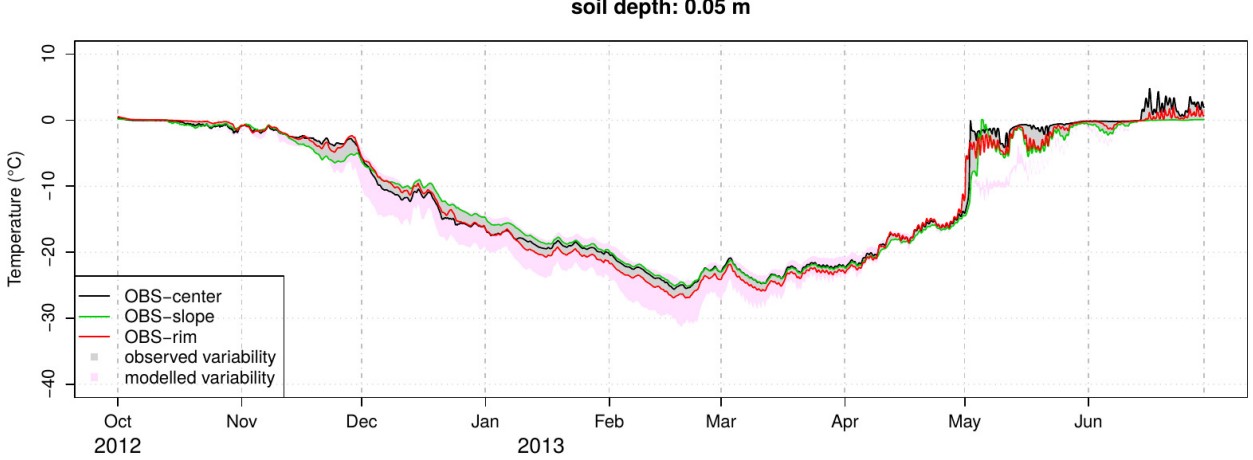

**Figure 11: Simulated and observed soil temperature variability (in °C) at 5 cm depth. Observed soil temperatures at rim, center and slope locations within the reference polygon are overlain in the colours shown.**



**Table 1: Nash-Sutcliff model efficiency criteria (Nash and Sutcliff, 1970) between the soil temperature simulations and measurements at different depths in the grass-center of the reference polygon.**

| Depth<br>setup | 5 cm | 20 cm | 40 cm |
|---|---|---|---|
| DEFAULT | 0.72 | 0.70 | 0.66 |
| WIND | 0.96 | 0.97 | 0.98 |
| WIND+VEG | 0.95 | 0.95 | 0.94 |
| WIND+VEG+ANISO | 0.96 | 0.97 | 0.97 |

5    **Table 2: End-of-season properties for DH and wind-slabs.**

| | DH | Wind-slabs |
|---|---|---|
| Density (kg m$^{-3}$) | 225 | 360 |
| $K_{eff\text{-}z}$ (W m$^{-1}$ K$^{-1}$) | 0.20 | 0.36 |





**Code availability**

The adaptations to SNOWPACK used in this study are not included in the SNOWPACK distribution but the description provided in the manuscript allows the simulations to be reproduced in their entirety.

**Data availability**

5  Meteorological and snow depth data are available at https://doi.org/10.1594/PANGAEA.879341.

**Appendices**

**Appendix A: Regression of anisotropy parameter $Q$ to grain size**

**A.1 Regression of $Q$ to optical diameter in data from Löwe et al. (2013)**

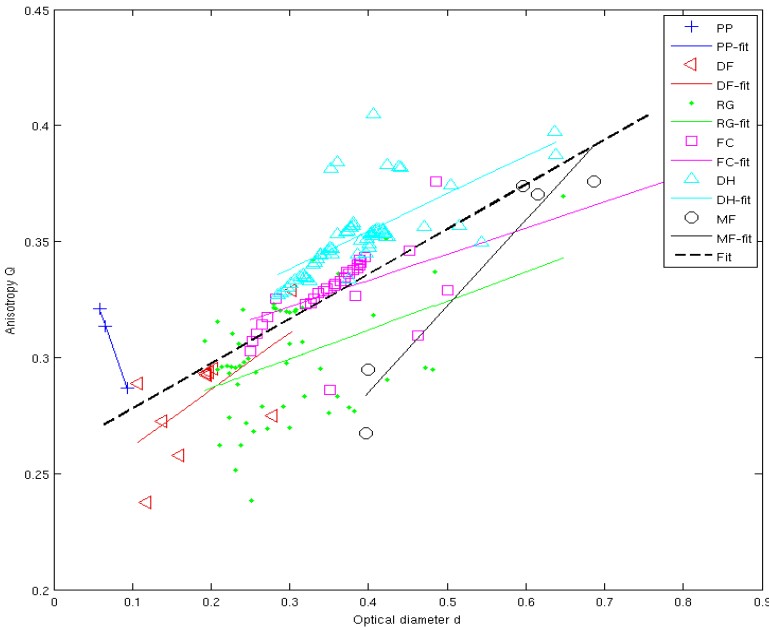

10  **Figure A1: Regression of anisotropy parameter $Q$ to optical diameter $d$ within snow type classes in data from Löwe et al. (2013).**



**Table A1: Regression coefficients for Fig. A1. All data within a snow type class were fitted to $Q^{REG}=a*d + b$, where $d$ is given in mm. When several grain types coexist, the dominant type is listed first.**

| Snow type | a [1/mm] | b [-] | $R^2$ |
|---|---|---|---|
| PP | -0.9631 | 0.3775 | 0.9954 |
| DF | 0.2450 | 0.2372 | 0.3981 |
| RG | 0.1250 | 0.2619 | 0.1872 |
| FC | 0.1132 | 0.2880 | 0.4356 |
| DH | 0.1620 | 0.2895 | 0.4645 |
| MF | 0.3733 | 0.1354 | 0.9155 |
| All | 0.1930 | 0.2587 | 0.4330 |

**A.2 Regression of $Q$ to SNOWPACK grain radius, used in the ANISO adaptation**

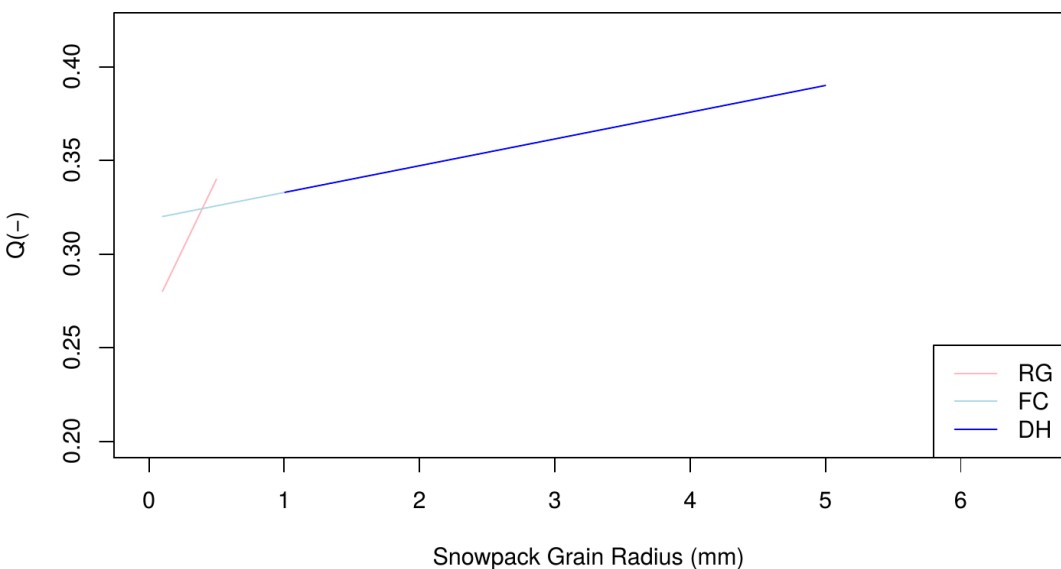

**Figure A2: Evolution of Q as parameterized in SNOWPACK ANISO adaptation.**

5    In the ANISO adaptation, $Q$ is parameterized as a function ($Q^{ANISO}$) of SNOWPACK grain radius ($rg$) for each of the FC, DH and RG snow type class:

$$Q^{ANISO}(rg) = \frac{Q^{REG}(d_{max})-Q^{REG}(d_{min})}{rg_{max}-rg_{min}} \cdot (rg - rg_{min}) + Q^{REG}(d_{min}) \qquad \text{(A1)}$$





where $rg_{max}$ and. $rg_{min}$ are the maximum and minimum values of $rg$ possibly achieved in SNOWPACK for the given snow type class (see Table A2), and $d_{max}$ and $d_{min}$ the maximum and minimum values of $d$ obtained in the data from Löwe et al. (2013) in the given snow type class.

Because SNOWPACK features a continuum between FC and DH grain radii, both grain type classes were merged in the ANISO adaptation by using and $Q^{REG}(d_{max})$ from DH and $Q^{REG}(d_{min})$ from FC in Eq. (A1) (see Fig. A2).



**Table A2: Parameters of the ANISO adaptation; Eq. (A1)**

| Snow type | $rg_{min}$ (mm) | $rg_{max}$ (mm) | $Q^{REG}(d_{min})$ | $Q^{REG}(d_{max})$ |
|---|---|---|---|---|
| RG | 0.1 | 0.5 | 0.28 | 0.34 |
| FC | 0.1 | 1 | 0.32 | |
| DH | 1 | 5 | | 0.39 |
| FC and DH | 0.1 | 5 | 0.32 | 0.39 |

**Appendix B: Construction of snow depth, DH height, $K_{eff-z}$ and $R_{th}$ time-series at the transect data points**

A visual estimate of the DH thickness and total snow depth was made at each of the 31 transect points (*pt*), based on the NIR image from date *t2*=2013-04-20 (estimated accuracy +/- 0.5 cm).

5 The following assumptions were made in the construction of DH thickness and snow depth (*HS(t)*) time-series over the entire snow season consistent with observations made at date *t2*:

- The snow depth was assumed to build up in a spatially homogeneous manner until date *t1*=2012-10-31 (confirmed by time-lapse photographs of the reference polygon). All 31 data points were therefore attributed the same snow depth until that date (i.e. the corrected snow depth ($HS_{50}(t)$ measured by the SR50 sensor).
10 Erosion-deposition processes subsequently lead to different accumulations ($HS_{pt}$) at each point along the transect. Do to the shortage of data, we linearly scaled $HS_{50}(t)$ that matched the end-of-season snow depth ($HS_{pt}(t2)$) for each point:

$$HS_{pt}(t > t1) = HS_{SR50}(t1) + \frac{HS_{pt}(t2) - HS_{SR50}(t1)}{HS(t2) - HS_{SR50}(t1)} \cdot (HS(t) - HS_{SR50}(t1)) \qquad (A2)$$

- We also considered a homogeneous DH build-up until *t1*: we used the DH build-up from the
15 WIND+VEG+ANISO simulation for all transect points until *t1*. For *t>t1*, we considered the DH thickness at each transect point to increase linearly to its end-of-season value. An exception was made when the observed end-of-season DH thickness was less than the modelled DH thickness at *t1*: in this case we considered the DH thickness to remain constant after its end-of-season value had been reached in the SNOWPACK simulation.

The constructed snow depth and DH thickness time series are illustrated in Fig. B1.



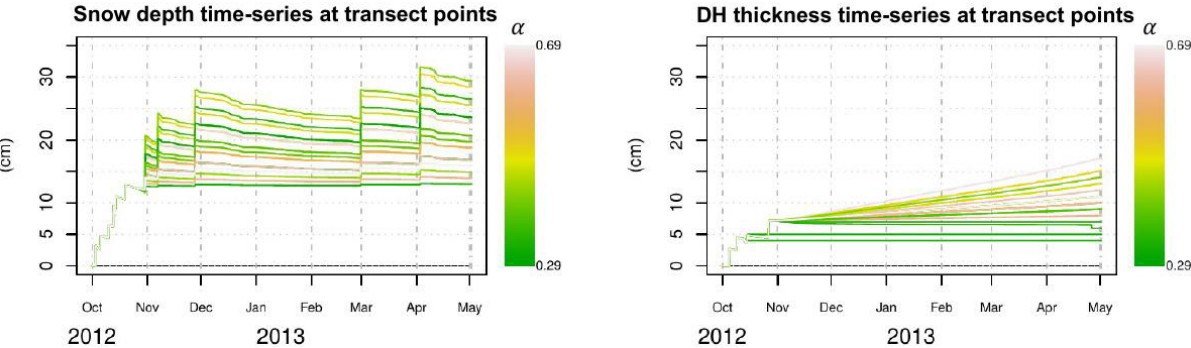

**Figure B1: Constructed snow depth and DH thickness time-series for each transect point. As in the manuscript, $\alpha$ is the DH-thickness to total snow depth ratio at time t2.**

Applying the 2-layer approach to the snow depths and DH thickness time-series using the snow properties described in the

5    text (Sect. 6) leads to the $K_{eff\text{-}z}$ and $R_{th}$ ensembles illustrated in Fig. B2.

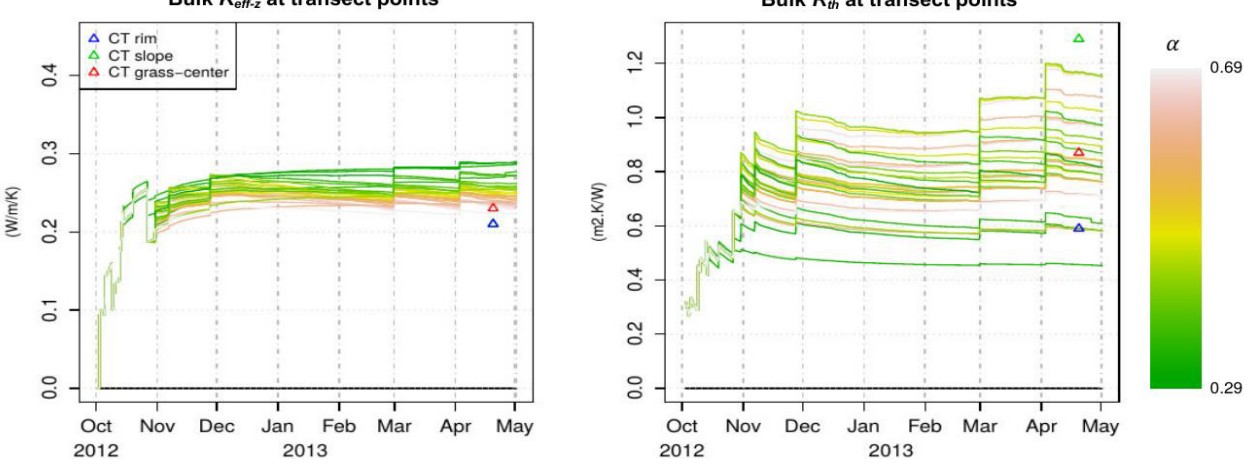

**Figure 12: Simulated $K_{eff\text{-}z}$ and $R_{th}$ time-series at the 31 transect data points. Overlain are the bulk properties estimated at the rim, slope and grass-center CT sites.**



## Appendix C: Thermal gradient between air and soil (5 cm depth)

**Temperature gradient through snow (K/m)**

**Figure C1: Temperature gradient between air and soil (5 cm depth) at the grass-center of the reference polygon.**

### Team Composition

5  Isabelle Gouttevin (IG), Moritz Langer (ML), Henning Löwe (HL), Julia Boike (JB), Martin Proksch (MP) and Martin Schneebeli (MS).

### Author contribution

M.S. and M.P. conceived the snow study. M.P. collected the snow data. M.L. and J.B. contributed the soil, permafrost and meteorological measurements. I.G., M.L., and H.L. performed the numerical simulations. All authors contributed to the
10  conception of the study. I.G. prepared the manuscript with contributions from all co-authors.

### Competing interests

The authors declare that they have no conflict of interest.



**Special issue statement**

This article is submitted to the joint special issue "Changing Permafrost in the Arctic and its Global Effects in the 21st Century (PAGE21)" (BG/ESSD/GMD/TC inter-journal SI).

**Acknowledgements**

We thank Mathias Bavay for his constant help and brilliant management of the SNOWPACK core structure and integration of new developments. Our sincere thanks go also to Michael Lehning and Charles Fierz for providing funding, enthusiasm, and insightful comments on development choices. We thank Florent Dominé for his vision of the processes at stake and the critical model limitations, and Jean-Emmanuel Sicart for very valuable comments. The field work for M.P. was supported by the INTERACT project SSTIS.

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
