# Peer review of "Observation and modelling of snow at a polygonal tundra permafrost site: spatial variability and thermal implications"

_The Cryosphere, 2017_

## Referee Comment (RC1) · Anonymous Referee #1 · 19 Jan 2018

**Observation and modelling of snow at a polygonal tundra permafrost site: spatial variability and thermal implications**

The study explores the spatial variability of snow properties at the polygonal tundra site at Samoylov site, Russia. Researchers use 1D snow and heat flow models to study spatial impact of snow on polygonal tundra, which somewhat limits the full understanding of the corresponding changes. The manuscript includes solid description of the current methods used to measure snow thermo-physical properties. The overall results states that adding the effect of wind, vegetation, and anisotropy improves the modeling of the ground temperatures at the Samoylov site. The current version of the paper lacks of discussion of the previous work that includes the effect of wind and vegetation (see SnowModel by Liston and Elder [2006]). It is important to include the description on how the SnowModel is different/similar from the current model development.

I was not sure what is the purpose of doing WIND, WIND+VEG, WIND+VEG+ANISO, since all three looks similar to me (Figure 3). Wind is the dominant factor in tundra, why should we care about other cases? For example, if you calculate the total difference between calculated and observed ground temperatures, I bet you would not see much improvement between those three cases. The changes in the snow over first part of the winter (dark winter) can be done by increasing snow density (i.e. chose the right empirical formula and adjust snow densities). To me the most interesting part would be matching temperatures toward the end of snow season (snowmelt). How should it be done, what kind of parametrization can improve the Figure 10, May jump in the temperatures.

Overall, there is a lack of the discussion on what scientific knowledge does it add to the current state of knowledge on snow. The flow in the manuscript require further improvement. Snow modeling literature review has to be complemented with the work by Liston and Elder (2006). How these results can be extrapolated locally/globally? What improvements the CLM modeling community have to do in order to improve snow representation in the current CLM type models? Please check and add that to the literature review. Current version of manuscript requires flow improvement and more clarity.

**Abstract**

P1. L20. Introduce the definition of the snow anisotropy.

P1. L23. Similarly, 'depth hoar' has to be defined.

P1. L24-25. "The potential of an …", this sentence is not clear to me.

P1. L25. "Dark part of winter" has to be defined.

P1. L25-26. Is that local to the Samoylov only?

P1. L27. It is common to reference Brown et al., (1997) about 24% of the land in Northern Hemisphere occupied by permafrost. Instead I suggest to say significant portion of land in Northern Hemisphere since permafrost is dynamic and shrinking spatially.

P2. L7. I would say, that soil temperatures beneath the thick snowpack would usually be warmer …. What is the difference between snowpack and seasonal snow?

P2.L9. Why to study snow in tundra is important? To me, in tundra vegetation should not play much role, I would say that the wind will play the most dominant effect on snow. Why would even consider the effect of vegetation?

-. L17. 'HS' change to $h_s$, otherwise it is associated with word abbreviation.

This paragraph has a lot of abbreviations (CT, HFP and so on). I suggest to make a table that reader can quickly refers to when forgets the abbreviation. The table can include short description of the method and a reference.

P3. Model literature review paragraph does not include work by Liston and Elder (2006), which includes wind and vegetation. What lessons can be learned from that model?

P3.L27. 1m high rims. Is that true?

P5.L26 Add the equation used for the heat conductivity.

P5.L29. Changes "figures" to "values". Does that mean that for other temperatures (not -10 ℃) the values will change? Do you know what is the possible range?

Figure 3A. Differentiating snow layers by colors are confusing, since several colors looks the same to me. Consider no colors, just boundaries to separate layers and add notations inside each layer (RG, FC and so on), can also increase the resolution to fit the notation.

Figure 3B. Bulk density and $K_{eff}$ are they step functions or piece-wise linear functions?

P9.L8. define the $R_{th}$

Figure 6. How anisotropy (Q) was calculated? Provide an equation.

Section 3.3. List equations for C2011, R2013, L2013. Why only these three formulas? How do they compare with Sturm et al., (1997) or Goodrich (1982) or others equations for conductivities? It is not clear how those empirical relationships account for Q?

P12. L12-15. How $K_{eff}$ is calculated for each 4 cases (DEFAULT, WIND, …)?

Figure 7. I assume these profiles are simulated by model. Are the grain type inputs or calculated by model? How these grain types evolve in the model?

Figure8. Make c and d plots. Separate rim from grass. Is that possible to plot snow observed texture next to the profiles?

P13. L23-26. Phases1-4 show them on the Figure 9.

Section 6. Think about how you can revise that section. There is too much information in it, which is hard to follow.

Figure 11. The colors on the plot is hard to see (especially magenta).

P18. L30. P19.L24. It will be interesting to discuss how new version CROCUS might change the result of current modeling. It looks to me that the conductive heat transport within the snowpack during snowmelt is complemented by an adjective heat transport that the melted snow water carries with it in the snowpack. Typically, the temperature gradient changes in sign or fluctuates near 0 C (making thermal conductivity useless during snowmelt). It will be nice to discuss what could be an easy (straight forward) way to parametrize the adjective heat transfer introduced by flowing water in the snowpack.

**References**

1. Liston, G.E., Elder, K., 2006. A distributed snow-evolution modeling system (SnowModel). J. Hydrometeorol. 7, 1259–1276. http://dx.doi.org/10.1175/ JHM548.1.

2.  Sturm, M., Holmgren, J., Konig, M., Morris, K., 1997. The thermal conductivity of seasonal snow. J. Glaciol. 43 (143).
3.  Goodrich, L.E., 1982. The influence of snow cover on the ground thermal regime. Can. Geotech. J. 19, 421–432.

---

## Referee Comment (RC2) · Anonymous Referee #2 · 30 Jan 2018

**Reviewer's Comments**

doi.org/10.5194/tc-2017-280

Observation and modelling of snow at a polygonal tundra permafrost site: spatial variability and thermal implications

Isabelle Gouttevin, Moritz Langer, Henning Löwe, Julia Boike, Martin Proksch, and Martin Schneebeli

**General comments:**

The paper highlights results from a field campaign carried out in order to assess the performance of the SNOWPACK model to simulate arctic snow conditions. A focus was made on the thermal conductivity and modifications to SNOWACK (wind compaction, vegetation and vapor flux) have shown improvements in snow simulations by SNOWPACK.

This paper address a major problem in arctic snow simulations where most models and not well adapted, which leads to significant biases in snow microstructure, which in turn creates problems for the radiative transfer community. For the past years, several efforts were made to improve and adapt snow models to arctic conditions, but the success was limited. This paper thus represent a major step forward, that will certainly help numerous scientific communities 9rmeote sensing, ecology, hydrology, …).

I recommend this paper for publication in The Cryosphere, after minor revisions detailed below.

**Specific comments:**

- Last paragraph of introduction: the paragraph simply describes the various section of the paper. Typically, such paragraph can be found in theses, but I think it is not relevant here. I would remove this paragraph, which would reduce the introduction (already quite long).

- Figure-1 should include coordinates.

- The use of NIR to calculate the ratio of DH with respect to snow depth should be more detailed. Photos are simply showed with explanation on the method used to distinguish DH. Was the calculation made automatically, or was a threshold applied on reflectance?

- Section 2.2.1: More details is needed regarding the spatial representativity of the SR50 measurements. Authors mention that small differences can be due to local scale variability, but a quantification should be done. Typically, the spatial variability is caught within 30-35m (1m spacing) in open tundra environments. What was the variability around the site? How does the average depths around the site compare to the SR50 measurements? This should be clarified, especially since SNOWPACK is forced on observed depths by the SR50 (section 2.4).

- Section 2.4: the authors are well aware of the sensitivity of SNOWPACK to uncertainties in meteorological forcing data. Many products exist, the authors should justify why using ERA-interim rather that other meteorological products… Also, it is mentioned that a comparison with in-situ meteorological stations showed that ERA is 'suitable'…this should be clarified.

- Page 10, last sentence. Can you please clarify that you adjusted only the VEG…and not VAP…so that VEG would account for VAP+VEG processes?

- Section 7.4.: there needs to be a discussion on meteorological forcing uncertainties… The resolution of ERA is quite large compared to a single site.

- On the pdf, the figures are general poor quality-resolution such as would be a simple printscreen. Please ensure high resolution on final version as some axis are hard to read

---

## Author Comment (AC1) · 28 Mar 2018

**Response to Comments by Anonymous Reviewers #1 and #2**

We sincerely thank the two anonymous reviewers for their constructive comments, and henceforth their substantial contribution to the improvement of the manuscript.

In this Response, Reviewers'comments are in blue, response to them is in black, modifications to the manuscript are highlighted in yellow. A corrected version of the manuscript, including Figures, is provided after the Response (from p. 18 on).

**Response to Reviewer #1**

**General comments**

1. **The study explores the spatial variability of snow properties at the polygonal tundra site at Samoylov site, Russia. Researchers use 1D snow and heat flow models to study spatial impact of snow on polygonal tundra, which somewhat limits the full understanding of the corresponding changes.**

We thank the reviewer for pointing our this important limitation, which we only partially mentionned and adressed in Sect. 7.4.

To day, only few 3-D thermo-hydrological models have been deployed over polygonal tundra landscapes. We are especially aware of recent work by Kumar et al. (2016), with the PFLOTRAN model deployed over 4 different polygonal tundra sites. At these sites, a considerable amount of field data were collected, which is currently unequalled eslewhere in the Arctic : meteorological data (including air temperature, summer precipitation, snow depth, relative humidity, wind speed and radiation), soil temperature data (at 16 different depths for 9 points across 4 transects, one at each site), and detailed subsurface characterization through 30 cm deep soil cores at each site and micro-topographic condition. Despite this tremendous amount of data, the authors *«believe that insufficient characterization and parameterization of heterogeneous properties [of the soil] due to limited data availability is one of the key reasons for [the warm model] bias. »* and further invoke *«poorly bounded [thermal] boundary conditions at the bottom of the modeling domain »*.

This means that the state-of-the art observations and models still fail to fully address the problem of spatial variability in soil temperatures 3-dimensionally.

Knowing these limitations, we addressed the question of the thermal impact of spatial variability in snow conditions from a 1-D perspective, which has 2 main limitations : (i) we omit the lateral heat conduction flux and (ii) we disregard the spatial variability in soil thermal properties and water content.

Despite these limitations, we believe our study has value, as it provides a needed complement to present-time studies like the one by Kumar et al. (2016), who focused only on soil processes. For a comprehensive (future) 3D model it is necessary that all factors contributing to spatial variability in soil temperatures, most notably snow, are well characterized and we regard our study as an important step in this direction.

Following the Reviewer's suggestion, we made the limitations of using a 1-dimensional model in Sect. 7.4 more explicit:

*« Our approach disregards the spatial variability in soil properties and soil saturation, which is also related to micro-topography, as well as the lateral heat fluxes between different landscape units. Distributed, 3-dimensional simulations that include the effect of snow redistribution by wind and spatial variations in soil conditions could, in theory, support a more consistent assessment of spatial variability in soil temperatures. However, they require a considerable amount of in situ data that is currently unavailable even at the most instrumented sites (Kumar et al., 2016). »*

**2.   The current version of the paper lacks of discussion of the previous work that includes the effect of wind and vegetation (see SnowModel by Liston and Elder [2006]). It is important to include the description on how the SnowModel is different/similar from the current model development.**

We thank the reviewer for pointing out the missing reference to the work of Liston and Elder (2006). Our approach in the treatment of wind effect on vegetion follows the work of Liston and Elder (2006) by considering no accumulation of high-density, wind-blown snow as long as snow depth has not exceeded the snow-holding capacity of the basal vegetation. This generates a lower density of the snowpack in a « vegetation layer », which actually corresponds to the snow-holding capacity defined by Liston and Elder (2006).

We clarified the relationship of our work to the work by Liston and Elder (2006) by stressing both the heritage and differences to this work in the manuscript :

**L2 p11 :** *« The underlying hypotheses are that i) while snow hasn't filled the snow-holding capacity of the basal vegetation, snow is not available for transport (Liston and Elder, 2006) and therefore snow accumulation in the grass-layer consists in precipitation particles of lower density than typical wind-blown rounded grains; and ii) that grasses form a rigid structure that protects snow from wind compaction. »*

**L7 p 11 :** *« Note that our approach however differs from the SnowModel by Liston and Elder (2006) in the sense that we focus on snow microstructure and properties (density, $K_{th}$) as influenced by the wind conditions, while the SnowModel and its blowing snow sublimation and redistribution scheme SnowTran3D target the spatial distribution and time evolution snow-water-equivalent, and the way they are affected by vegetation. »*

**3.  I was not sure what is the purpose of doing WIND, WIND+VEG, WIND+VEG+ANISO, since all three looks similar to me (Figure 3). Wind is the dominant factor in tundra, why should we care about other cases? For example, if you calculate the total difference between calculated and observed ground temperatures, I bet you would not see much improvement between those three cases. The changes in the snow over first part of the winter (dark winter) can be done by increasing snow density (i.e. chose the right empirical formula and adjust snow densities).**

We assume that the Reviewer is referring to Figure 7 in this comments, as Figure 1 to 6 of the original manuscript do not deal with WIND, WIND+VEG and WIND+VEG+ANISO simulations. In Figure 9 and 10, we indeed inter-compared the **simulated soil temperatures** in the WIND, WIND+VEG and WIND+VEG+ANISO setups, and compared them to observations. Figure 10 (which shows an improved setup with respect to Figure 9, see original manuscript) shows equal performances over winter phases 1 to 3 for WIND and WIND+VEG, while WIND+VEG+ANISO performs better.

However, soil temperature is not the only benchmark variable that we considered, as a proper modelling of the snow stratigraphy is essential for other important applications of SNOWPACK-like models. To improve the consistency of the paper and justify our effort to simulate a reasonable snowpack structure, we added the following lines to the Introduction :

5 *« A reliable simulation of snow structure in SNOWPACK-like models is essential not only for the simulation of the ground thermal regime but also for a variety of applications ranging from the exploitation of remote-sensing data (e.g. Montpetit et al., 2013), to the assessment of snowpack structure impact on wildlife (e.g. Ouellet et al., 2017). »*

Following this philosophy, Figures 7 and 8 clearly reveal that the WIND option alone is not enough to capture snow vertical structure in terms of stratigraphy, density and $K_{eff-z}$ profile : whichever empirical « wind-density » formulation you adjust, if it takes only wind into account, it will fail to reproduce the vertical structure of these variables. Oppositely, the WIND+VEG 10 setups perform better in this respect, while not degrading the bulk values of these variables.

In conclusion, wind is surely the dominant factor shaping the snowpack in tundra environments, but basal vegetation (here the high sedges) introduces a second-order modulation by i) affecting the surface roughness at the soil-atmosphere interface (snow-retention effect of grasses) and ii) affecting the snow compaction and snow metamorphic processes. It is important that snow models deployed in the Arctic for a variety of applications are able to account for these processes.

**4.   To me the most interesting part would be matching temperatures toward the end of snow season (snowmelt). How should it be done, what kind of parametrization can improve the Figure 10, May jump in the temperatures.**

We completely agree with the high interest of studying melt-time soil temperature and associated processes. However, these were clearly not the target of our study, which focuses on dry snow properties and their thermal impact, as outlined by the 20 INTERACT SSTIS project that provided our funding. Therefore, our modelling strategy did not target the melt period, and our model design is not appropriate to study the associated processes.

**5.   Overall, there is a lack of the discussion on what scientific knowledge does it add to the current state of knowledge on snow.**

25 We believe that our contributions to the state of the knowledge on Arctic snow are thouroughly summarized in the Conclusion of our manuscript as follows :

- 1. An analysis of the drivers of the spatial variability in snow insulating power across a polygonal tundra landscape, and an assessment of the thermal impact of this variability
- 2. First CT estimates of DH and wind-slab conductivities in an Arctic tundra snowpack (validated against the 30 ground thermal regime through numerical simulations)
- 3. An inter-comparison between 3 recent parameterizations for $K_{eff-z}$ with different philosophies (see later in the Response to specific comments)
- 4. The sucessful use of a density and anisotropy-based parameterization of $K_{eff-z}$ in a  detailed snow model (as envisionned by Löwe et al., 2013)
35 - 5. The highlight of the early and polar night winter periods as the most sensitive to the thermal properties of snow, helping prioritize the future snow investigations through field work or modelling

These points are respectively discussed in Sections 3.2, 7.1, 3.3, 5.1 and 7.1, 7.3.

If we missed to raise other relevant points, we would appreciate if the reviewer could more specifically pinpoint what is missing.

**6. The flow in the manuscript require further improvement / Current version of manuscript requires flow improvement and more clarity.**

We worked on several aspects to improve the flow and clarity of the manuscript :

- First, the objectives and decsription of the paper's approach was refined at the end of the Introduction, following joint advice from Reviewer #2 :

*« Our objectives in this study were (1) to investigate the thermal properties of snow in an Arctic snowpack and their link to microstructure and microtopography, (2) propose adaptations to a detailed snow model to these local snow conditions, to be validated against snow and soil temperature observations and (3) quantify the thermal impact of spatial variability in snow depth and snow structure across a typical polygonal tundra microtopography. To this end we relied on snow samples analysed by CT, on a variety of in situ snow observations collected during a dedicated field program at Samoylov in April 2013, and on more long-term observations on meteorology and soil variables. The model adaptations we propose were made to the detailed snow model SNOWPACK, which we used in combination with the CryoGrid3 (CG3, Westermann et al., 2016) permafrost-soil model for the simulation of the ground thermal regime, as this model was extensively validated at Samoylov. »*

- Second, section 3 and section 6 were completely revised, including condensation of the text, paragraph merging and structural changes. Typically, the focus of Section 3 was tightened around the thermal properties of the snowpack. This is reflected by changes in the titles and subtitles, in the organisation of the paragraphs and in the shortening of the section. Section 6 was mostly re-written and simplified. We do not exhibit here all the changes made, and refer to the manuscript with corrections highlighted for an overview of the modifications.
- Finally, effort was made to better synthesize relevant informations in the Figures, leading to one Figure less in the revised manuscript.

**7. Snow modeling literature review has to be complemented with the work by Liston and Elder (2006).**

Done – see response to 2.

**8. How these results can be extrapolated locally/globally?**

More field data originating from other tundra environment would definetly be ideal to further validate and generalize parts of our findings. This is typically the case for the characterization of the spatial variability of the snow thermal properties, and for the evaluation of the modified SNOWPACK simulations on soil temperature observations. These data would ideally feature co-located soil and surface temperature observations, and snow-depth sensors, at places with intensive characterization of the soil (many replication of measurements within each micro-topographical class) to exclude or limit error compensations originating from soil thermal modelling.

However, some of our findings have a more general value, as the highlight of the early and polar night winter periods as the most sensitive to the thermal properties of snow, or the inter-comparison of the 3 parameterization for $K_{eff-z}$.

We complemented our manuscript so as to specify the local vs general flavour of our conclusions:

- Sect 7.1 *: « On the one hand, our adaptations to SNOWPACK are inherently local, tied to the specific Samoylov conditions, and should be verified at other tundra sites comprising co-located snow and soil observations together with a complete set of meteorological driving data. »*
- We recall a sentence from the Conclusion : « *We also estimated the impact of the natural snowpack spatial variability on the underlying permafrost thermal regime during an entire winter, based on our $K_{eff-z}$ and density observations and on our understanding of the snowpack dynamics. Beyond this quantitative estimate, which is intrinsically tied to the local climatology and micro-topography of our site, an important conclusion is that the sensitivity of the ground thermal regime to the overlying snow reaches a maximum during the cooling winter period, when temperature gradients between atmosphere and soil are at their steepest.* »

**9. What improvements the CLM modeling community have to do in order to improve snow representation in the current CLM type models? Please check and add that to the literature review.**

We answer here regarding the possibilities of improvement of the snow schemes of GCMs and ESM in tundra environement only (excluding schrub tundra and taiga). From our study, we judge that significant improvements would proceed from (i) taking wind compaction into account, (ii) representing a basal DH layer (of lower density / conductivity), and (iii) carefully considering the extinction of solar radiation within very dense snowpacks. To our knowledge, this is not yet the case in most snow schemes (e.g. Wang et al., 2013). Furthermore, these snow schemes should be validated at places with co-located observations of snow (depth, SWE), meteorological variables and soil temperatures, that help verify that the mass balance, density and thermal role of the snowpack are captured properly. This latter point is raised in Sect 7.4 of the original manuscript. To enrich our manuscript with the former elements, we complemented the Conclusion section:

« *Finally, our study pinpointed processes that exert an important control on the ground thermal regime of tundra regions while being neglected in the snow schemes of general circulation models or earth system models (e.g. Wang et al., 2013): the effect of wind compaction and DH growth on the insulating power of tundra snow, as well as the enhanced extinction of solar radiations in by dense wind-crusts within the snowpack. This suggests possible ways to improve snow modelling over Arctic regions in these models, of benefit for the representation of permafrost -processes.* »

**Abstract**

**P1. L20. Introduce the definition of the snow anisotropy.**

We here refer to the structural anisotropy of the three-dimensional microstructure, or loosely, the anisotropy of « Snow grains and arrangements thereof », meaning that they do not have the same properties in all directions. We added « *microstructural* » to the abstract to point this out. The definition of anisotropy with regard to $K_{eff-z}$ is given in the Introduction at the first occurrence of this term.

**P1. L23. Similarly, 'depth hoar' has to be defined.**

Depth hoar is one of the 9 main grain shaped identified by the International Classification for seasonal Snow on the Ground (Fierz et al., 2009) and very common to snow observers. We do not feel it is necessary to explicitly repeat the definition from the classification.

**P1. L24-25. "The potential of an ...", this sentence is not clear to me.**

We changed this sentence in the abstract, which should now be clearer :

« *Also, a density and anisotropy-based parameterization for $K_{eff-z}$ lead to further slight improvements* »

**P1. L25. "Dark part of winter" has to be defined.**

We changed this sentence in the abstract :

« *Soil temperatures were found to be particularly sensitive to snow conditions during the early winter and polar night, highlighting the need for improved snow characterization and modelling over this period.* »

**P1. L25-26. Is that local to the Samoylov only?**

It is likely the case in most of the Arctic land surface area experiencing low Snow accumulation (<50 cm), as this sensitivity is driven by the high value of the temperature gradient between the (warm) soil and (cold) air, which is at its highest during this period and particularly strong where the snowpack is shallow. The general value of this statement is stressed in the Conclusion (see point 8 above).

**P1. L27. It is common to reference Brown et al., (1997) about 24% of the land in Northern Hemisphere occupied by permafrost. Instead I suggest to say significant portion of land in Northern Hemisphere since permafrost is dynamic and shrinking spatially.**

We thank the reviewer for this correction, and added the referece in the manuscript. However, we decided to propose a rough approximation of the exact estimate from the cited publication (23.9 %), for permafrost is not the focus of our paper and this approximation is just meant to propose an order of magnitude to the reader. In this respect, the approximation we propose is - to our knowledge - still valid : mean global temperature change between 1998 and now is around 0.2 °C, and permafrost sensitivity to global warming has been assessed to $4 \pm 1 \cdot 10^6$ km$^2$ C° $^{-1}$ (Chadburn et al., 2017), hence a change of less than 1% of permafrost extent is to be expected between 1998 and now.

**P2. L7. I would say, that soil temperatures beneath the thick snowpack would usually be warmer .... What is the difference between snowpack and seasonal snow?**

*Snow* is the material / medium, while *snowpack* refers to the snow covering a land surface. Snow can be grown/analysed in laboratory, where it does not form a snowpack anymore.

In the present study, « snowpack » and « Snow » can be interchanged most of the time, because we study in situ snow, within a snowpack. Both are seasonal, because the snow cover at Samoylov does not last all year long. Elsewhere like in Antarctica or in glaciers'accumulation zone, snow or snowpack could be perenial.

**P2.L9. Why to study snow in tundra is important? To me, in tundra vegetation should not play much role, I would say that the wind will play the most dominant effect on snow. Why would even consider the effect of vegetation?**

See response to point 3 above.

**-. L17. 'HS' change to hs, otherwise it is associated with word abbreviation.**

HS is the official abbreviation for Height of Snowpack / Snow depth in the International Classification for seasonal Snow on the Ground (Fierz et al., 2009). Following your concern, we were careful to use only the official terms when designing this variable, changing one occurrence of « snow height » to « snow depth » in the manuscript.

**This paragraph has a lot of abbreviations (CT, HFP and so on). I suggest to make a table that reader can quickly refers to when forgets the abbreviation. The table can include short description of the method and a reference.**

We generally agree about curse and blessing of acronyms. We think though that the amount of acronyms in the present manuscript is still average and prefer to keep tables for actual findings from our study.

**P3. Model literature review paragraph does not include work by Liston and Elder (2006), which includes wind and vegetation. What lessons can be learned from that model?**

See response to point 2 above.

**P3.L27. 1m high rims. Is that true?**

The height of the rims was indeed over-estimated in this statement.

We corrected the manuscript based on 3D laser-scanning performed in 2017 :

« […] rims that are 20 to 50 cm high ».

**P5.L26 Add the equation used for the heat conductivity.**

The calculation of the effective thermal conductivity is based on the solution of the stationary (pore scale) heat equation which is solved directly on the binary CT image. This is now explicitly stated in Sect. 2.2.2.

**P5.L29. Changes "figures" to "values". Does that mean that for other temperatures (not -10 oC) the values will change? Do you know what is the possible range?**

Done. Air conductivity diminishes and ice conductivity increases when temperature decreases. Typical values (in W m$^{-1}$ K$^{-1}$) are given in the following Table from engineering toolbox.com :

|  | 0°C | -10°C | -20°C | -30°C |
|---|---|---|---|---|
| ki | 2.22 | 2.30 | 2.39 | 2.50 |
| ka | 0.024 | 0.024 | 0.023 | 0.022 |
| ki/ka | 92.5 | 95.8 | 103.9 | 113.6 |

The effective thermal conductivity of snow inherits the temperature dependence from both, ice and air conductivity in a non-linear way. This effect of temperature could be estimated from the parametrization given in Löwe et al. (2013) which incorporates it via the explicit dependence of the parametrization on the phase conductivities. For an order of magnitude, the respective values for 0°C and -30°C given above can be used : we obtain a relative difference of 7 % for an isotropic microstructure with ice volume fraction of 0.3. The (ki, ka) values used for the analysis in the paper lie in the middle of that range, leading to an uncertainty of less than 4 %.

We amended the manuscript to include this estimation :

P5 L28 : « These values approximate the conductivity of the air and ice medium at temperatures between -15 °C and -20 °C (cf. engineering toolbox.com and data compiled by Waite et al., 2006), causing a maximum error in retrieved $K_{eff}$ of less than 4 % for a snowpack between 0°C and -40°C (estimation based on the parametrization from Löwe et al 2013 using the respective values of ka and ki). »

**Figure 3A. Differentiating snow layers by colors are confusing, since several colors looks the same to me. Consider no colors, just boundaries to separate layers and add notations inside each layer (RG, FC and so on), can also increase the resolution to fit the notation.**

We changed the plot and preferentially used symbols to represent grain shapes, instead of colours.

**Figure 3B. Bulk density and Keff are they step functions or piece-wise linear functions?**

Bulk density is one single value per profile, indicated as text over each profile's plot in Fig. 3b.

Density and $K_{eff-z}$ profiles show one (averaged) value over each small volume or depth interval where the measurement or estimation was actually made. This depth interval depends on the instrument or estimation methods. It is typically around 2.5 cm for the CT estimates of $K_{eff-z}$ and density shown in Fig. 3b. The snow profiles were not continuously sampled for density and $K_{eff-z}$ with the CT method. Instead, a few (4 to 6) depths intervals of about 2.5 cm, were selected, and the CT analysis provided density and $K_{eff-z}$ estimates averaged over these depth intervals (plain segments in Fig. 3b). For visualization purposes, a dashed line connects these plain segments in Fig. 3b, but it does not represent any measurement.

We clarify this feature in the caption of Fig. 3 :

« Density and $K_{eff-z}$ values are represented by piecewise constant functions over the layers where the CT analysis was performed; these segments are connected by a dashed line as a guide to the eye.

**P9.L8. define the Rth**

Done.

**Figure 6. How anisotropy (Q) was calculated? Provide an equation.**

Providing the equation for Q would require a considerable overhead in additional, technical details from an entire section in Löwe et al. (2013), where this parameter was introduced. Reproducing these explanations in our paper, would considerably reduce its focus and introduce unecessary complexity. We agree though that a non-technical description of Q is required in the manuscript, so we clarified the link between Q and micro-structure, and directly refered to the equation in Löwe et al. (2013).

« *It relies on an anisotropy parameter, Q, calculated directly from CT images* ==based on the two-point correlation function (Löwe et al., 2013, their Eq. 4)== »

**Section 3.3. List equations for C2011, R2013, L2013. Why only these three formulas? How do they compare with Sturm et al., (1997) or Goodrich (1982) or others equations for conductivities? It is not clear how those empirical relationships account for Q?**

Various comparisons of different parametrizations already exist in literature. The reason to include just these three is the following :

Sturm et al. (1997) and C2011 rely on the exact same methodology, i.e. squared regression to density. As C2011 re-assessed the formulation from Sturm et al. (1997) based on recent data using the NP technique, it felt consistent to use C2011 as a benchmark for CT-assessed $K_{eff\text{-}z}$. Then, R2013 and L2013 both differ from the philosophy of Sturm et al. (1997) and C2011, by regressing against anisotropic grain types with vertical preferential direction (R2013) and by relying on theoretical bounds for $K_{eff\text{-}z}$ (L2013). From a comparison of these three it is possible to reveal the uncertainties in effective thermal conductivity for a snowpack (that is dominated by anisotropic depth hoar) when using a generic density-parametrization (C2011), or a density-parametrization that was mostly derived from anisotropic snow (R2013), or L2013 that explicitly accounts for both, density and anisotropy effects for a depth hoar dominated tundra snowpack.

**P12. L12-15. How Keff is calculated for each 4 cases (DEFAULT, WIND, ...)?**

We better explained this in the manuscript :

Sect. 4 : ==« *All setups except the one including the ANISO adaptation rely on the original $K_{eff\text{-}z}$ parameterization from SNOWPACK described in Sect. 2.5.* »==

**Figure 7. I assume these profiles are simulated by model. Are the grain type inputs or calculated by model? How these grain types evolve in the model?**

The grain type are simulated by the SNOWPACK model. We add this element to the description of SNOWPACK in section 2.5, where references to the snow grain evolution laws are given.

Sect. 2.5 : *« Snow is represented by a number of state variables (temperature, density, and water content) and the snow micro-structure by grain characteristics (grain size, size of bonds, sphericity, and dendricity)* ==*which allow a diagnostic of the*==

5   ==*grain type (Lehning et al., 2002b).* »==

**Figure8. Make c and d plots. Separate rim from grass. Is that possible to plot snow observed texture next to the profiles?**

This concerns Fig. 7 now. We added the observed snow shape and suppressed the ice-center profile which notably differed

10  in terms of involved processes.

**P13. L23-26. Phases1-4 show them on the Figure 9.**

Done, also on the Fig. 10 and 11 of the original manuscript.

15  **Section 6. Think about how you can revise that section. There is too much information in it, which is hard to follow.**

See response to point 6.

**Figure 11. The colors on the plot is hard to see (especially magenta).**

20  Magenta color was changed to lightblue, improving the plot's readability.

**P18. L30. P19.L24. It will be interesting to discuss how new version CROCUS might change the result of current modeling. It looks to me that the conductive heat transport within the snowpack during snowmelt is complemented by an adjective heat transport that the melted snow water carries with it in the snowpack. Typically, the temperature**

25  **gradient changes in sign or fluctuates near 0 C (making thermal conductivity useless during snowmelt). It will be nice to discuss what could be an easy (straight forward) way to parametrize the adjective heat transfer introduced by flowing water in the snowpack.**

We interpret that the Reviewer means « advected heat transport ».

The very interesting question raised by the Reviewer's comment is however out of the scope of our manuscript, as explained

30  in  point 4.

Last paragraph of introduction: the paragraph simply describes the various section of the paper. Typically, such

5    paragraph can be found in theses, but I think it is not relevant here. I would remove this paragraph, which would reduce the introduction (already quite long).

Done.

Figure-1 should include coordinates.

10   Done.

The use of NIR to calculate the ratio of DH with respect to snow depth should be more detailed. Photos are simply showed with explanation on the method used to distinguish DH. Was the calculation made automatically, or was a threshold applied on reflectance?

15   Following the reviewer's advice we now provide more details as to the treatment of the NIR images in the Methods section :

Sect 2.2.1: *Near-infrared (NIR) images of the trench were realized to characterize the thickness of the basal depth hoar (DH) layer along this transect at 50 cm spatial steps.* ==*The NIR-images were treated in ImageJ (Schneider et al., 2012) by the following procedure: the green channel was extracted from the RGB-image. The brightness and contrast was visually optimised based on the histogram. The average brightness of the full profile was 125, the depth hoar region 106, the surface*==

20   ==*layer 125 (brightness range 0-255). The boundary between these two main layers was measured based on a ruler put in the center of the image. The resolution of the NIR images was better than 0.1 mm, so depth hoar crystals and especially depth hoar chains were in addition easy to discriminate from the upper layer with smaller, mostly rounded grains*==.

Section 2.2.1: More details is needed regarding the spatial representativity of the SR50 measurements. Authors

25   mention that small differences can be due to local scale variability, but a quantification should be done. Typically, the spatial variability is caught within 30-35m (1m spacing) in open tundra environments.

What was the variability around the site? How does the average depths around the site compare to the SR50 measurements? This should be clarified, especially since SNOWPACK is forced on observed depths by the SR50 (section 2.4).

30   The SR50 is placed 1.23 m above snow-free ground with a beam angle of approximately 22°. The SR50 measure is therefore representative of a circle of radius (surface) ranging from 0.23 m ($0.17m^2$) in snow-free conditions, to 0.19 m ($0.12 m^2$) with 20 cm of snow.

Spatial variability of the snowpack was indeed not investigated in our manuscript at such small, decimeter scale. However, we can rely on two sources to assess that this small-scale spatial variability in snow depth can be large. First, snow depth and

DH height were recorded at 0.2 m steps along a grass-polygon transect nearby the Samoylov station (Fig. R1). Although the delimitation of slope, rim and center is approximate and observer-dependant, these data illustrate that snow depth variations can reach 10 cm over 20 cm distance at Samoylov, which is half the footprint of the SR50.

[Figure]

5 **Figure R1: Snow (lightblue) and DH height (dark blue) along a 13.6 m transect in a grass-polygon in the vicinity of the Samoylov main station, on 19-04-2013. An estimation of the local micro-topography is also given.**

Second, spatial variability of snow depth at the decimeter scale in link to (vegetation-induced) micro-relief was investigated by earlier publications, and we especially recall transect data from Sturm and Holgren (1994) showing spatial variations even higher than recorded at Samoylov over the sedge tussocks tundra landscape at Imnavait Creek, Alaska (Fig. R2, extracted

10 from Sturm and Holgren (1994)).

[Figure]

Fig. 2. A cross section through the snow cover at Imnavait Creek, 18 November 1989. Depth hoar is indicated by an inverted "V"; wind slab is indicated by a dot with a line through it. Note discontinuous snow strata that pinch out over tussock tops. In the lower part of the figure, the temperature of the snow/ground interface is shown.

**Figure R2. Extract from Sturm and Holgren, 1994.**

For our SNOWPACK simulations we made use of SR50 data, rescaled so that their value in late April 2013 (13 cm) matched the mean of observed Snow depth at 20 polygon centers sites at this date (20 cm). This difference of 7 cm between SR50 Snow depth and the mean of observed snow depths at the manual snowpits from center locations, can well be explained by the small, decimeter scale variability in snow depth, as illustrated above.

We thank the reviewer for pointing out this missing quantification, and amended the manuscript subsequently :

*P5 L12 : Snow depth was recorded continuously over the 2012-2013 snow season by an SR50 sensor (Campbell Scientific, ± 1 cm accuracy, ± 1 cm precision) located in the topographically low center of the reference polygon (Fig. 1). This instrument acquires data over a circular surface of ~ 20 cm radius. However, this snow depth record differed from data acquired at grass-center snowpits: on 21 April 2013 the SR50 measured 13 cm of snow while both the transect, CT and snowpit data indicated depths in excess of 17 cm for grass-center conditions (Fig. 3). This difference is likely due to small scale variability in snow depth induced by micro-relief (notably vegetation tussocks) and in processes such as wind erosion immediately below the SR50 sensor: ancillary snow depth data acquired over a 14 m grass polygon transect at 20 cm spatial resolution show a 7 cm variance in snow depth, and variations up to 9 cm over 40 cm horizontal distance in center conditions. To build a representative snow depth record for grass-center conditions, we matched the SR50 snow data to the median of manually recorded snow depths at grass-center snowpits (20 cm) on 21 April 2013, by multiplying the SR50 record by a constant factor of 1.6. The 7 cm offset in late April is consistent given the observed small-scale variability in snow depth. Finally, a time-lapse camera provided daily, low-resolution images of the reference polygon.*

**Section 2.4: the authors are well aware of the sensitivity of SNOWPACK to uncertainties in meteorological forcing data. Many products exist, the authors should justify why using ERA-interim rather that other meteorological**

**products... Also, it is mentioned that a comparison with in-situ meteorological stations showed that ERA is 'suitable'...this should be clarified.**

**Section 7.4.: there needs to be a discussion on meteorological forcing uncertainties... The resolution of ERA is quite large compared to a single site.**

5 The only ERAi data used in our study were wind-speed, radiation, plus air temperatures over a 6-week period in February-March. Here we rely on the following elements to justify the use of this atmospheric product for our application :

   i) Temperature, radiation and wind fields are rather homogeneous over large spatial scales in this Northern Sibirian region in the absence of marked relief perturbing the synoptic western atmospheric flow (Brun et al., 2013).

10   ii) Following, the local values of these atmospheric variables at Samoylov should not depart much from the reanalyzed field over larger (80 km for ERAi) scales

   iii) An exception of (ii) occurs for air temperature where oceans/seas cover a significant part of the reanalysis grid-cell. However, the Samoylov Island in the Lena delta is far enough from the Laptev sea coast, for the entire ERA-i grid-cell containing Samoylov to be considered as an inland pixel in the reanalysis.

15   iv) A comparison between air temperature observed at Samoylov and the reanalysis field for the 2012-2013 snow season (excepting 1-02-2013 to 15-03-2013 when Samoylov sensor saturated), confirms the extreme good quality of the ERA-i product for air temperatures (Fig R3).

   v) ERA-i was shown by previous literature to be an adequate forcing for Snow simulations in North-Eastern Europe including Siberia (Brun et al., 2013).

20   vi) Finally, precipitation is often the atmospheric variable the most mis-represented in realyses fields (Troy et al., 2011), but first, ERA-i only only minorly suffers from this issue  for winter precipitation (Troy et al., 2011 ; Brun et al. 2013) and second, we circumvent this possible issue by forcing the SNOWPACK model with observed Snow depths.

[Figure]

**Figure R3: Comparison between ERA-i 2m air temperature (x axis) and 2m air temperature measured at the Samoylov station (y axis) over 2012-2013 (excluding the 1-02-2013 to 15-03-2013 period). Green line is the linear regression between these values.**

Following the Reviewer's advice we modified the manuscript as follows :

5  **Sect 2.4:** *Unfortunately the sensor (HMP45, Campbell Scientific) became saturated at temperatures below -40 °C and so for the period between 1 February and 15 March 2013, when the air temperatures were below -40 °C, we used air temperature records from the ERA-interim reanalysis (ERA-i; Dee et al., 2011) instead:* ==for the rest of the 2012-2013 winter period, ERA-I temperatures show a high correlation with Samoylov observations (r²=0.97) and a low bias (-0.9°C).== *The incoming shortwave and longwave radiation and the wind-speed were also taken from ERA-i as none of these variables was recorded*

10  *at Samoylov during the 2012-2013 snow season.* ==ERA-I fields were proven to be a high quality source of driving variables to simulate the evolution of the Nothern Eurasian snowpack including Sibiria (Brun et al., 2013), with minor differences between station data and grid-field over large, rather flat areas like the Lena Delta.== *A comparison of ERA-i with locally acquired meteorological data* ==from earlier years at Samoylov furthermore confirmed this validity for the skin surface temperature, which responds very sensitively to differences in the driving variables (Langer et al., 2013).==

15  **Sect. 7.4:** *This uncertainty,* ==together with uncertainty in the meteorological forcing that cannot be completely excluded,== *also affects our estimates of the thermal impact of snow spatial variability. Continuous monitoring of ice depletion at the base of*

*the snowpack, and snow monitoring programs focusing on the early and dark winter periods at sites comprising both meteorological and radiation observations, would help to provide better constraints for the thermal characteristics of the snowpack and the underlying metamorphic processes at this time, yielding substantial benefits for the next generation of coupled snow-soil models.*

5 **Page 10, last sentence. Can you please clarify that you adjusted only the VEG...and not VAP...so that VEG would account for VAP+VEG processes?**

Indeed. We amended the manuscript for more clarity :

*« We therefore chose to address both VAP and VEG together: both effects are comprised in the phenomenological "VEG" adaptation, described below. »*

**On the pdf, the figures are general poor quality-resolution such as would be a simple printscreen. Please ensure high resolution on final version as some axis are hard to read.**

Figures were carefully checked and modified, taking into account recommandations from Reviewer #1. Fig. 7 from the original manuscript, now Fig. 6, was notably improved.

[revised manuscript text omitted]